# Metabolic activities are selective modulators for individual segmentation clock processes

Mitsuhiro Matsuda [1,2] ✉, Jorge Lázaro [1,3] & Miki Ebisuya [1,2,4] ✉

Numerous cellular and molecular processes during embryonic development prompt the fundamental question of how their tempos are coordinated and whether a common global modulator exists. While the segmentation clock tempo scales with the kinetics of gene expression and degradation processes of the core clock gene *Hes7* across mammals, the coordination of these processes remains unclear. This study examines whether metabolic activities serve as a global modulator for the segmentation clock, finding them to be selective instead. Several metabolic inhibitions extend the clock period but affect key processes differently: glycolysis inhibition slows Hes7 protein degradation and production delay without altering intron delay, while electron transport chain inhibition extends intron delay without influencing the other processes. Combinations of distinct metabolic inhibitions exhibit synergistic effects. We propose that the scaled kinetics of segmentation clock processes across species may result from combined selective modulators shaped by evolutionary constraints, rather than a single global modulator.

During the progression of embryonic development, multiple cellular and molecular processes happen sequentially and concurrently: while cells undergo proliferation, differentiation, and intercellular communication, a myriad of gene products are synthesized, processed, transported, and degraded in each cell. Since altering the timing or duration in one process without adjusting that in the others can perturb the precise spatiotemporal patterning of cells and tissues, coordinating tempo across multiple processes is crucial. This prompts the question of whether a common global factor exists to modulate the tempo of these processes simultaneously. Moreover, given that developmental tempo varies across animal species, another fundamental question is whether different species utilize the same global modulator.

There has been a debate on the existence and nature of a global modulator of developmental tempo[1–3]. For example, temperature can be a global modulator for ectotherms, such as insects, fish, and frogs[4–9]. As most biochemical reactions accelerate or decelerate by 2–3 folds for every 10-degree change in temperature, the external temperature can synchronously set the tempo of most cellular and molecular processes. By contrast, endotherms including mammals develop at relatively constant, similar temperatures, and thus these animals are unlikely to use temperature as a global modulator. Another attractive candidate for a global modulator is metabolism or energy. As all biological processes require energy, alteration in metabolism may influence multiple processes simultaneously. Indeed, mitochondrial activity is associated with several molecular and cellular processes, including gene expression, protein degradation, and neuronal development[10–13]. At the organismal level, nutrition is well-known to influence growth, and allometric metabolic rates are proposed to underlie diverse growth rates across species[14,15]. Mutants for mitochondrial components also display sluggish development in C. elegans and mice[16–19]. However, it remains unclear to what extent metabolism can modulate the tempo of multiple processes simultaneously.

The segmentation clock is widely used as a model to study developmental tempo. The oscillatory gene expression of the segmentation clock regulates the timings of periodic body segment formation during embryogenesis[20]. The segmentation clock has been recapitulated from pluripotent stem cells of multiple species by several groups, and the oscillation periods are reproducible despite variations in the differentiation protocols and cell lines[21–28]. The in vitro

¹European Molecular Biology Laboratory (EMBL) Barcelona, Barcelona, Spain. ²Cluster of Excellence Physics of Life, TU Dresden, Dresden, Germany. ³Collaboration for joint PhD degree between EMBL and Heidelberg University, Faculty of Biosciences, Heidelberg, Germany. ⁴Max Planck Institute of Molecular Cell Biology and Genetics, Dresden, Germany. ✉e-mail: mitsuhiro.matsuda@tu-dresden.de; miki.ebisuya@tu-dresden.de

segmentation clock is amenable to detailed measurements of the kinetics of molecular processes. The core molecular mechanism of segmentation clock oscillation is a delayed negative feedback loop of the *Hes7* gene (Fig. 1a): the Hes7 protein produced through gene expression processes represses its own promoter, before being degraded eventually. The negative feedback loop with delays can give rise to oscillatory gene expression of Hes7, and its oscillation period is mainly determined by the degradation rates and delays in the loop[29–31]; slower degradation and longer delays lead to a longer period. A previous study demonstrated that the protein/mRNA degradation rates,

the production delay (i.e., the delay derived from production processes such as transcription and translation), and the intron delay (i.e., the delay associated with intron sequences such as splicing) of the *Hes7* gene could largely account for the cell-autonomous periods of mouse and human segmentation clocks (Fig. 1a)[25].

The segmentation clock periods are notably species-specific (Fig. 1b); even among mammals, the periods are 2–3 h in mice and rabbits, ~4 h in rhinoceroses, pigs, and cattle, ~5 h in humans, and ~6 h in common marmosets[20,22,24,25,27,28]. The three key kinetic parameters of the *Hes7* gene, namely the protein degradation rate, intron delay, and

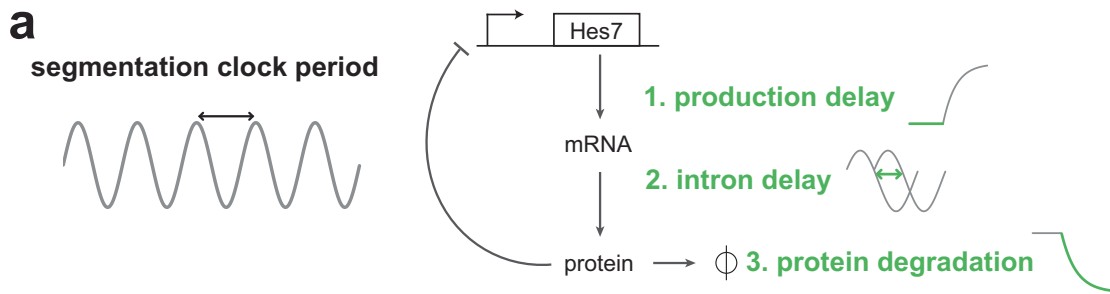

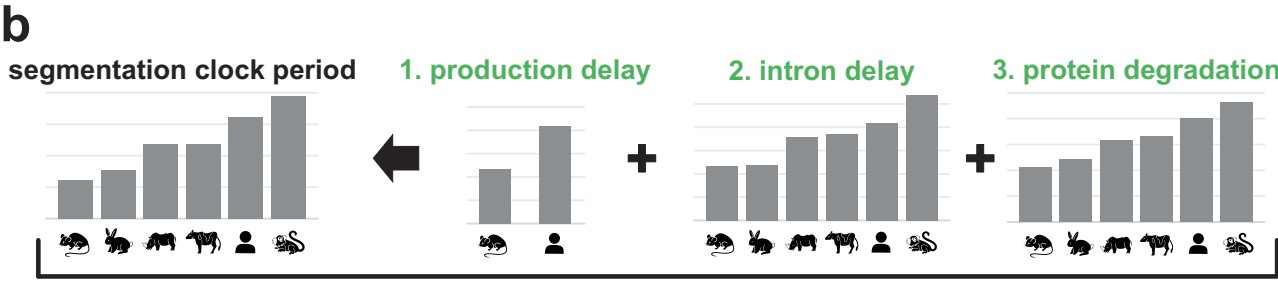

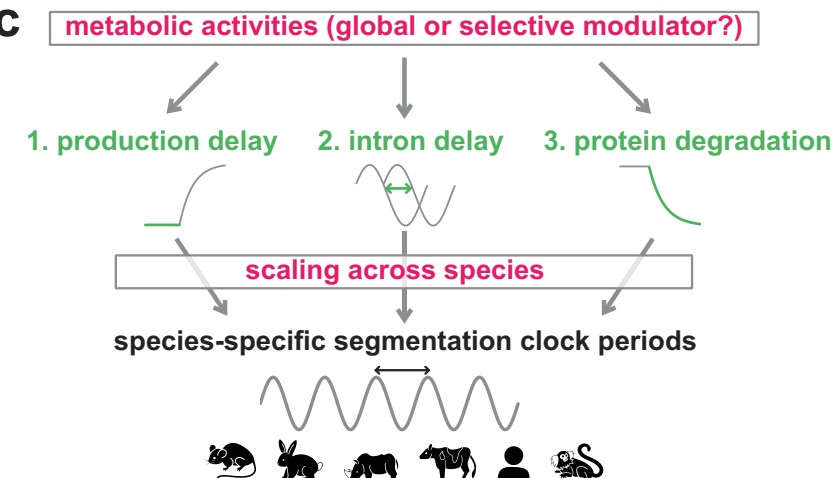

**Fig. 1 | Scaling of the segmentation clock and the question. a** Schematic representation of the core mechanism of segmentation clock oscillation: a delayed negative feedback loop of the *Hes7* gene. Hes7 is a transcriptional repressor that inhibits its own promoter, giving rise to the oscillatory expression. The production process of the Hes7 protein takes time due to transcription and translation (1. production delay) whereas the Hes7 intron processing also takes time (2. intron delay). The Hes7 protein is eventually degraded through the ubiquitin-proteasome pathway (3. protein degradation). The oscillation period, namely the segmentation clock period, is largely determined by the degradation rates of Hes7 mRNA/protein as well as the total delay in Hes7 gene expression (the sum of the production delay

and intron delay). Hes7 mRNA degradation rate is not directly assessed in this study. **b** The key kinetic parameters of Hes7 (1. production delay, 2. intron delay, and 3. protein degradation rate) are highly correlated with the segmentation clock period across mouse, rabbit, rhinoceros, bovine, human, and marmoset cells. This proportional change in the parameters is termed scaling in this study. Data for the graphs and animal icons are from Matsuda et al.[25] and Lázaro et al.[27]. The production delay data are available only for mouse and human. **c** This study addresses whether metabolic activities act as a global modulator that simultaneously affects the three key molecular processes of the segmentation clock or selective modulators.

production delay, are highly correlated with the segmentation clock periods across species (Fig. 1b)[25,27]. Although it is theoretically possible, for example, to extend the period by extending only the intron delay without altering the other two kinetic parameters, each species simultaneously and proportionally modulates the three parameters to give rise to the species-specific period. The remarkable scaling of these parameters implies but does not prove, the existence of a common global modulator that simultaneously controls the protein degradation, intron processing, and production processes of Hes7 across species. Alternatively, these individual molecular processes may have been modulated by separate mechanisms through evolutionary selection (Fig. 1c).

Metabolism is known to affect the segmentation clock. Modulation of glycolytic activity disturbs somitogenesis in embryos[32–36] whereas modulation of electron transport chain (ETC) activity can alter the period of the in vitro segmentation clock[26,27]. These results raise a hypothesis that metabolism may be a global modulator that simultaneously controls the multiple molecular processes of Hes7 to tune the segmentation clock period (Fig. 1c). It must be noted, however, that metabolism is an umbrella term representing a broad spectrum of activities and phenomena, including ATP production, ETC activity, glycolysis, mTOR pathway signaling, metabolites, protein/RNA turnover, and even mitochondrial morphology. Indeed, recent studies reported that rather than ATP production, more specific metabolic factors, such as the cytosolic NAD + /NADH ratio, intracellular pH, and glycolytic metabolites, are crucial for the segmentation clock and somitogenesis[26,34–36]. As the metabolism hypothesis originally gained popularity because of the universal role of energy and ATP in biological processes, it is debatable if a specific metabolic activity can act as a global modulator for the key molecular processes of the segmentation clock (Fig. 1c).

In this study, we impose different types of metabolic inhibitions on the segmentation clock. Unlike temperature change, the effects of metabolic inhibitions on the three key molecular processes of Hes7 are selective, casting doubt on the concept of metabolic activities as a global modulator for the segmentation clock tempo.

## Results
### Metabolic inhibitions selectively affect clock processes
We examined the effects of different types of metabolic inhibitors on the mouse segmentation clock period (Fig. 2a). An ETC inhibitor, sodium azide (azide) is reported to decelerate the segmentation clock[26,27], and a glycolysis inhibitor, 2-Deoxy-D-glucose (2DG) is reported to perturb somitogenesis[33,34]. Mouse and human presomitic mesoderm (PSM) cells were induced from mouse Epiblast stem cells (EpiSCs) and human induced pluripotent stem cells (iPSCs), respectively, and the segmentation clock oscillation in PSM cells was visualized with the Hes7 promoter-luciferase reporter[23,25,27]. Glycolysis inhibition or ETC inhibition extended the mouse segmentation clock period in a dose-dependent manner (Fig. 2b, c: Supplementary Fig. 1). The periods in mouse PSM cells treated with 10 mM 2DG and 1 mM azide were $191 \pm 2.1$ min (mean ± sd) and $204 \pm 8.8$ min, respectively. While they were significantly longer than the control mouse period ($152 \pm 3.5$ min), they were still much shorter than the human period ($325 \pm 5.5$ min).

We then examined the effects of metabolic inhibitions on the three key kinetic parameters of the Hes7 gene that mostly accounted for the segmentation clock period: the Hes7 protein degradation rate, intron delay, and production delay (Fig. 1b)[25]. Protein degradation was measured by halting the transcription of the Hes7-Nanoluciferase (NLuc) reporter and monitoring the decay in the luciferase signal (Supplementary Fig. 2a)[25]. Glycolysis inhibition decelerated Hes7 protein degradation (Fig. 2d, e; Supplementary Fig. 2b); the Hes7 half-life measured in the 2DG-treated mouse PSM cells was $32 \pm 2.1$ min while that in untreated cells was $20 \pm 0.6$ min. We recently observed this

decelerating effect of 2DG on the degradation of numerous other proteins as a consistent trend[37]. By contrast, ETC inhibition by azide treatment did not show a significant influence on Hes7 protein degradation (Fig. 2d, e; Supplementary Fig. 2b). The intron delay represents a combined delay caused by the Hes7 intron sequence, including the delay due to intron splicing. The intron delay was defined as the oscillation phase difference between two Hes7 promoter-luciferase reporters with and without suences; the oscillation phase of the reporter without introns should always precede the one with introns (Fig. 2f; Supplementary Fig. 3)[25]. Azide treatment notably extended Hes7 intron delay; $20 \pm 2.0$ min in azide-treated mouse PSM cells as compared to $9 \pm 1.2$ min in untreated cells (Fig. 2g; Supplementary Fig. 4). By contrast, 2DG treatment did not show a significant influence on Hes7 intron delay. The production delay represents a combined delay caused by the gene expression steps of Hes7, including transcription and translation, except for intron-related steps. The production delay was measured by inducing the expression of the Hes7-NLuc reporter and monitoring the onset timing[25] (Supplementary Fig. 5a). While 2DG treatment extended Hes7 production delay ($21 \pm 1.1$ min in 2DG-treated mouse PSM cells as compared to $16 \pm 1.0$ min in untreated cells), azide treatment did not show a significant influence (Fig. 2h, i; Supplementary Fig. 5b). These results demonstrated selective and disproportionate effects of glycolysis and ETC inhibitions on the three key molecular processes of the mouse segmentation clock even though the two metabolic inhibitors extended the period to a similar extent; glycolysis inhibition selectively affected Hes7 protein degradation and production delay whereas ETC inhibition affected only Hes7 intron delay.

### Combined inhibitions synergistically extend the clock period
The selective and complementary effects of glycolysis and ETC inhibitions prompted us to investigate whether their combined inhibition could yield synergistic effects on the segmentation clock period. While individual treatments with either 2DG or azide led to only a modest extension of the period, their combinations notably extended it (Fig. 3a–d; Supplementary Fig. 6). For example, the period of mouse PSM cells treated with 1 mM azide and 2 mM 2DG was $287 \pm 13$ min, resembling the human period more closely than the mouse (Fig. 3b). Even at lower concentrations (0.3 mM azide and 1 mM 2DG), each of which individually showed only negligible effects (Fig. 2c), the combined treatment resulted in a clearly extended period of $206 \pm 0.7$ min (Fig. 3d). These observations led us to hypothesize that even a metabolic inhibitor previously deemed ineffective on the segmentation clock might exhibit a notable impact when combined with another inhibitor. Oligomycin is an ATP synthase inhibitor (Fig. 2a), and it was reported to have no significant influence on the segmentation clock period[26]. As high concentrations of oligomycin cause the dampening of segmentation clock oscillation[26], we employed low concentrations (10-100 nM). While oligomycin treatment alone exhibited a significant yet very modest extension of the period (e.g., $177 \pm 1.5$ min in mouse PSM cell treated with 10 nM oligomycin), combinations of oligomycin and 2DG notably extended it (e.g., $248 \pm 2.9$ min in mouse PSM cells treated with 1 mM 2DG + 10 nM oligomycin) (Fig. 3e, f). These results indicated synergistic effects of different types of metabolic inhibitions on the segmentation clock period. It must be noted, however, that metabolic inhibitions, especially 2DG and oligomycin treatment, lowered the reporter signal (Supplementary Fig. 1a; 6) and that prolonged exposure to combined metabolic inhibitions often resulted in cell death (Supplementary Fig. 1c).

To assess the effects of the different metabolic inhibitors on the cellular metabolic rates, we measured the oxygen consumption rate (OCR) and glycolytic ATP production rate (glycoATP) (Fig. 3g; Supplementary Fig. 7). The OCR and glycoATP were used as indicators for mitochondrial respiration and glycolytic rates, respectively. Treatment with the ETC inhibitor azide lowered the OCR without influencing the

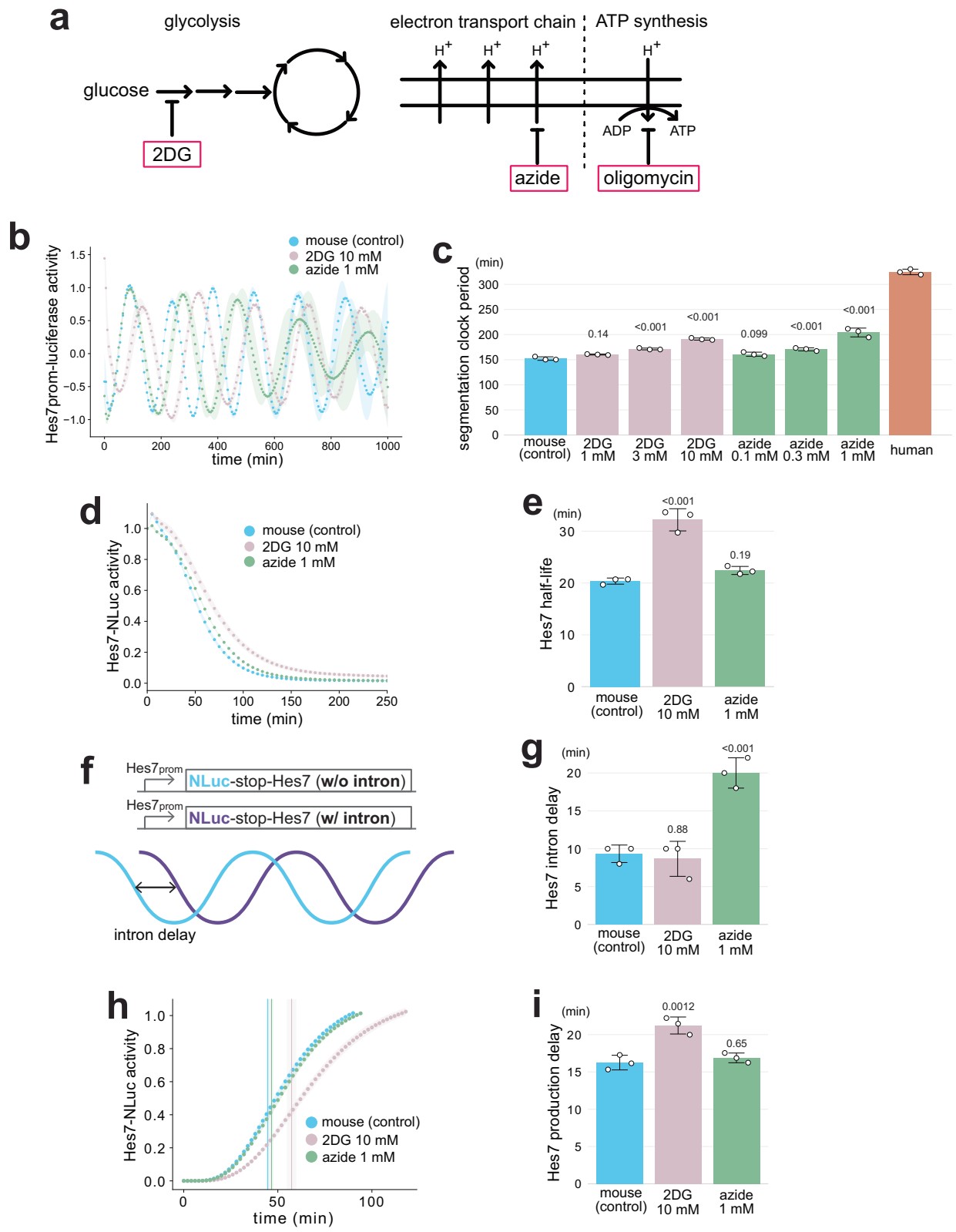

glycoATP (Fig. 3g), as expected. By contrast, the glycolysis inhibitor 2DG lowered the glycoATP and slightly upregulated the OCR, potentially due to a metabolic compensation mechanism (Fig. 3g; 10 mM 2DG). Combined treatment with a low concentration of 2DG and azide repressed the OCR to the same extent as azide treatment alone and did not significantly influence the glycoATP (Fig. 3g; 1 mM 2DG + 1 mM azide). In other words, the combined treatment did not exhibit any

detectable synergistic effect on the metabolic rates, even though the same combination (1 mM 2DG + 1 mM azide) synergistically extended the segmentation clock period (Fig. 3b, d). These results confirmed the efficacy of the metabolic inhibitors and suggested that the segmentation clock period did not directly correlate with the bio-energetic rates, at least with the OCR or glycoATP, consistent with the previous report[26].

**Fig. 2 | Selective effects of metabolic inhibitions on the key molecular processes of the segmentation clock. a** Schematic representation of cellular metabolic pathways and their pharmacological inhibitors. 2-Deoxy-D-glucose (2DG), sodium azide (azide), and oligomycin are known inhibitors of glycolysis, electron transport chain (ETC), and ATP synthesis, respectively. **b** Dose-dependent effects of 2DG or azide on the segmentation clock period. Mouse PSM cells were treated with the metabolic inhibitors from minus 2 h, and the oscillatory activity of the *Hes7* promoter-luciferase reporter was monitored. The signal was detrended and amplitude-normalized. **c** Hes7 oscillation periods estimated from **b**. Human data are from Fig. 5. **d** Effects of 2DG or azide on Hes7 protein degradation. Mouse PSM cells were treated with the inhibitors from minus 4 h. The transcription of the Hes7 reporter fused with Nano luciferase (NLuc) was halted upon the addition of doxycycline (Dox) at time 0, and the decay of the Hes7-NLuc signal was monitored. **e** Hes7 half-lives estimated from **d** and Supplementary Fig. 2. **f** Hes7 intron delay assay. The oscillation phase difference between the luciferase reporter without (w/o) *Hes7* intron sequences and the one with (w/) *Hes7* intron sequences was defined as the intron delay. **g** Effects of 2DG or azide on Hes7 intron delay. Mouse PSM cells were treated with the inhibitors from minus 2 h, and the oscillatory activities of the luciferase reporters w/o and w/ *Hes7* intron sequences were monitored. Intron delays were estimated from Supplementary Fig. 4. **h** Effects of 2DG or azide on Hes7 production delay. Mouse PSM cells were treated with the inhibitors from minus 4 h. The transcription of Hes7-NLuc was induced upon Dox addition at time 0, and the onset of the Hes7-NLuc signal was monitored. Vertical lines indicate the inflection points. **i,** Hes7 production delays estimated from **h** and Supplementary Fig. 5. **b, d, h** Shading indicates mean ± sd ($n = 3$). **c, e, g, i** Graphs indicate mean ± sd ($n = 3$). *P*-values are from two-sided Dunnett's test against the indicated controls. Source data are provided as a Source Data file.

Another metabolic indicator that has been associated with the segmentation clock is the cellular NAD + /NADH redox state. The ETC inhibitor azide has been reported to decrease the NAD + /NADH ratio and decelerate the segmentation clock[26]. To explore this relationship further, we measured the NAD + /NADH ratio under various metabolic inhibition conditions (Fig. 3h). Azide treatment indeed led to a slight, though not statistically significant, decrease in the NAD + /NADH ratio. Importantly, while the glycolysis inhibitor 2DG, like azide, slowed down the segmentation clock, it significantly increased the NAD + /NADH ratio. Among the five different metabolic inhibition conditions, including combined treatments, we could not find a correlation between the NAD + /NADH ratio and the segmentation clock period or any of the three kinetic parameters of Hes7. These results suggest that bioenergetics or redox state alone may not directly determine the segmentation clock period.

## Combined inhibitions simultaneously affect clock processes

The notable and synergistic effects of the combined metabolic inhibitions on the segmentation clock period prompted us to investigate their impact on the three key kinetic parameters of Hes7. The combined treatments with 2DG and azide, or 2DG and oligomycin, synergistically decelerated Hes7 protein degradation (Fig. 4a–d; Supplementary Fig. 8). For example, the Hes7 half-life in mouse PSM cells treated with 1 mM 2DG and 1 mM azide (32 ± 1.6 min) was clearly longer than that in untreated cells, while the treatment with the low concentration of 2DG or azide alone did not significantly influence the protein degradation (Fig. 4b). The treatment with 10 nM oligomycin and 1 mM 2DG further extended the Hes7 half-life to 67 ± 4.2 min (Fig. 4d). By contrast, the impact of the combined treatment with 2DG and azide on Hes7 intron delay (19 ± 2.3 min) was similar to that of the azide treatment alone (20 ± 2.0 min in Fig. 2g), and the combination of oligomycin and 2DG did not significantly influence the intron delay (Fig. 4e; Supplementary Fig. 9). The combined treatment with 2DG and azide synergistically extended Hes7 production delay (21 ± 2.4 min; Fig. 4f, g; Supplementary Fig. 10a), while the combination of oligomycin and 2DG showed no significant effect on the production delay (Fig. 4h, i; Supplementary Fig. 10b). These results indicated that combined metabolic inhibitions could yield synergistic effects, especially on Hes7 protein degradation and production delay. It is noteworthy that the combination of glycolysis and ETC inhibitions affected the three key molecular processes simultaneously (1 mM 2DG + 1 mM azide in Fig. 4), fulfilling the criteria as a global modulator for the segmentation clock tempo. This combined treatment with 2DG and azide achieved the most substantial period extension in both experimental data and simulations based on the three key kinetic parameters (Supplementary Fig. 10c).

## Metabolic inhibitions selectively affect human processes

We have so far demonstrated the selective effects of distinct metabolic inhibitions on the three key kinetic parameters of the segmentation clock in mouse PSM cells. To shed light on the scaling mechanisms of these parameters across animal species (Fig. 1c), we tested if the human segmentation clock processes show similarly selective responses to metabolic inhibitions. Glycolysis inhibition by 2DG treatment or ETC inhibition by azide treatment extended the human segmentation clock period (Fig. 5a, b; Supplementary Fig. 11a), and their combined treatment exhibited synergistic effects (Fig. 5c, d; Supplementary Fig. 11b). The HES7 protein degradation in human PSM cells was decelerated by 2DG, whereas it was not significantly influenced by azide (Fig. 5e, f; Supplementary Fig. 12). These effects of metabolic inhibitions on the human segmentation clock were consistent with those on the mouse clock. Note, however, that the impacts of 10 mM 2DG on the human segmentation clock appeared slightly milder than those on the mouse clock (compare Fig. 5b, f to Fig. 2c, e). The HES7 intron delay in human PSM cells was notably extended by azide but not by 2DG (Fig. 5g; Supplementary Fig. 13). The HES7 production delay in human PSM cells was extended by 2DG (Fig. 5h, i; Supplementary Fig. 14). Although the production delay was also extended by azide, the extension was negligible. These results obtained in human PSM cells were essentially consistent with the results from mouse PSM cells where 2DG predominantly affected Hes7 protein degradation and production delay whereas azide targeted Hes7 intron delay. As expected, 2DG treatment inhibited glycoATP whereas azide inhibited OCR in human PSM cells (Fig. 5j; Supplementary Fig. 15). Collectively, we concluded that individual metabolic inhibitions selectively affected the segmentation clock processes in multiple animal species.

## Temperature is a global modulator for clock processes

To contrast the selective modulation by metabolic inhibitions, we employed a temperature shift, a well-established method to modulate the segmentation clock in ectotherms[6,7,38]. In zebrafish, for example, the segmentation clock period changes more than 3 fold across the physiological developmental temperature range[7]: 18.7 min at 30.8°C and 55.4 min at 20.0°C. Consistent with this, we found that decreasing the culture temperature of mouse PSM cells from the standard 37 °C to 30 °C extended the mouse segmentation clock period (330 ± 3.8 min), mimicking well the human period (Fig. 6a, b; Supplementary Fig. 16a). We then examined the effects of the temperature decrease on the three key kinetic parameters of Hes7. Decreasing the temperature decelerated Hes7 protein degradation in mouse PSM cells (Hes7 half-life: 41 ± 2.4 min; Fig. 6c, d; Supplementary Fig. 16b). Moreover, the temperature decrease extended both Hes7 intron delay (35 ± 1.2 min; Fig. 6e; Supplementary Fig. 16c–e) and production delay (24 ± 1.2 min; Fig. 6f, g; Supplementary Fig. 16f). These results indicated that temperature could act as a global modulator that controls the three key molecular processes simultaneously. However, mice and humans should not have a large difference (such as 7 °C) in their body temperatures, and the mouse and human PSM cells cultured in vitro at the same temperature still show differential clock periods. Thus,

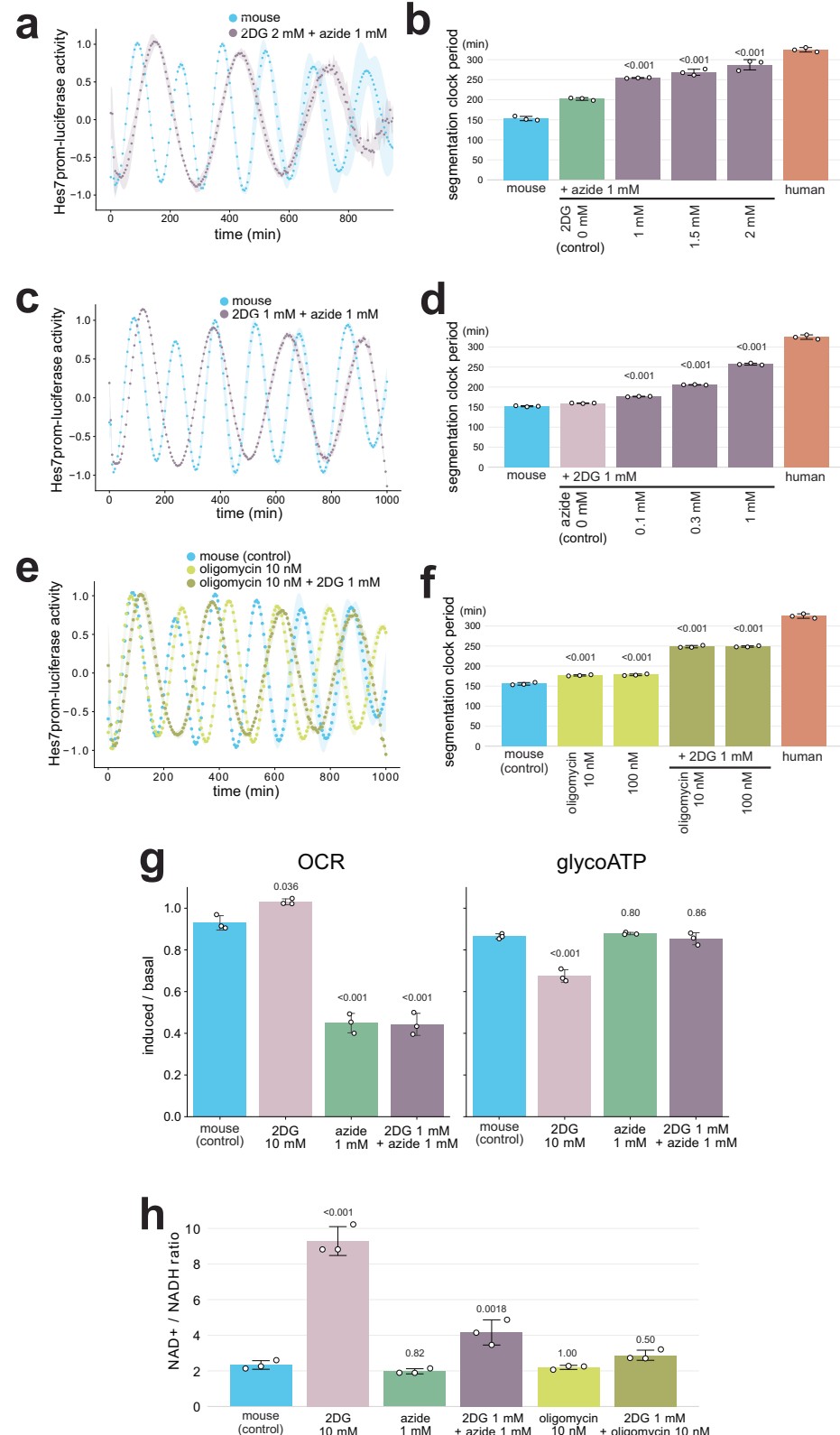

**Fig. 3 | Synergistic effects of metabolic inhibitions on the segmentation clock period. a, c, e** Synergistic effects of 2DG, azide, and oligomycin on the segmentation clock period. Mouse PSM cells were treated with the inhibitors from minus 4 h, and the oscillatory activity of the *Hes7* promoter-luciferase reporter was monitored. The signal was detrended and amplitude-normalized. Shading indicates mean ± sd (*n* = 3). **b, d, f** Hes7 oscillation periods estimated from **a, c, e**. Human data

are from Fig. 5. **g** Oxygen consumption rate (OCR) and glycolytic ATP production rate (glycoATP) measured by the Seahorse analyzer. The ratio of induced rates (after adding inhibitors) to basal rates (before adding inhibitors) was calculated from Supplementary Fig. 7. **h** Cellular NAD + /NADH ratio. **b, d, f, g, h** Graphs indicate mean ± sd (*n* = 3). *P*-values are from two-sided Dunnett's test against the indicated controls. Source data are provided as a Source Data file.

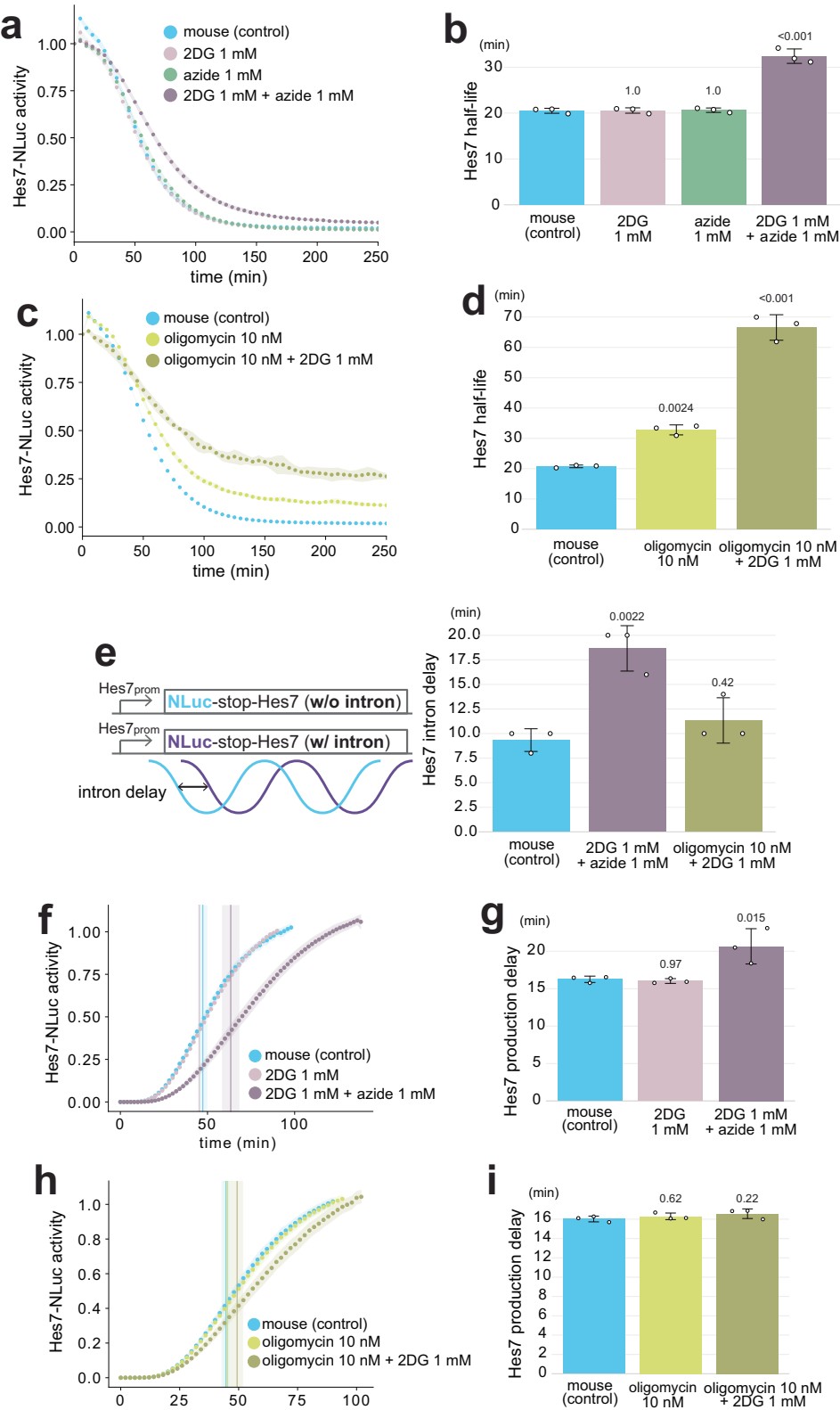

**Fig. 4 | Global effects of combined metabolic inhibitions on segmentation clock processes. a**, **c** Effects of 2DG, azide, and oligomycin on Hes7 protein degradation. Mouse PSM cells were treated with the inhibitors from minus 4 h, and the decay of the Hes7-NLuc signal was monitored. **b**, **d** Hes7 half-lives estimated from **a**, **c** and Supplementary Fig. 8. **e** Effects of 2DG, azide, and oligomycin on Hes7 intron delay. Mouse PSM cells were treated with the inhibitors from minus 4 h, and the oscillatory activities of the luciferase reporters w/o and w/ *Hes7* intron sequences were monitored. The intron delays were estimated from Supplementary Fig. 9.

**f**, **h** Effects of 2DG, azide, and oligomycin on Hes7 production delay. Mouse PSM cells were treated with the inhibitors from minus 4 h, and the onset of the Hes7-NLuc signal was monitored. Vertical lines indicate the inflection points. **g**, **i** Hes7 production delays estimated from **f**, **h** and Supplementary Fig. 10. **a**, **c**, **f**, **h** Shading indicates mean ± sd ($n = 3$). **b**, **d**, **e**, **g**, **i** Graphs indicate mean ± sd ($n = 3$). *P*-values are from two-sided Dunnett's test against the indicated controls. Source data are provided as a Source Data file.

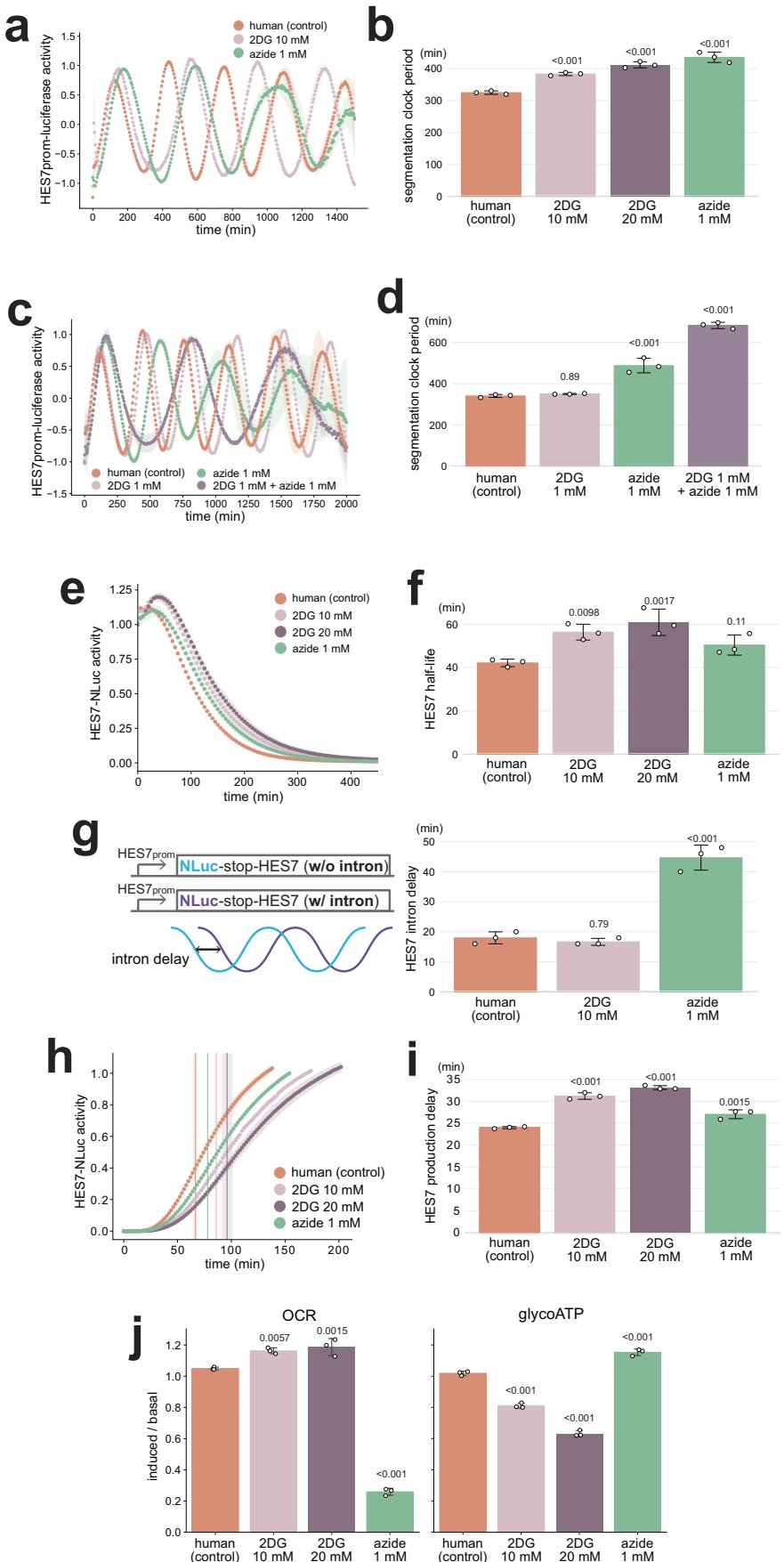

**Fig. 5 | Selective effects of metabolic inhibitions on the human segmentation clock. a, c** Effects of 2DG and azide on the human segmentation clock period. Human PSM cells were treated with the inhibitors from time 0, and the oscillatory activity of the *HES7* promoter-luciferase reporter was monitored. The signal was detrended and amplitude-normalized. **b, d** HES7 oscillation periods estimated from **a**. **e** Effects of 2DG or azide on HES7 protein degradation. Human PSM cells were treated with the inhibitors from minus 5 h, and the decay of the HES7-NLuc signal was monitored. **f** HES7 half-lives estimated from **e** and Supplementary Fig. 12. **g** Effects of 2DG or azide on HES7 intron delay. Human PSM cells were treated with the inhibitors from time 0, and the oscillatory activities of the luciferase reporters w/o and w/ *HES7* intron

sequences were monitored. The intron delays were estimated from Supplementary Fig. 13. **h** Effects of 2DG or azide on HES7 production delay. Human PSM cells were treated with the inhibitors from minus 4 h, and the onset of the HES7-NLuc signal was monitored. Vertical lines indicate the inflection points. **i** HES7 production delays estimated from **h** and Supplementary Fig. 14. **j** Oxygen consumption rate (OCR) and glycolytic ATP production rate (glycoATP) measured by the Seahorse analyzer. The ratio of induced rates (after adding inhibitors) to basal rates (before adding inhibitors) was calculated from Supplementary Fig. 15. **a, c, e, h** Shading indicates mean ± sd ($n = 3$). **b, d, f, g, i, j** Graphs indicate mean ± sd ($n = 3$). *P*-values are from two-sided Dunnett's test against the indicated controls. Source data are provided as a Source Data file.

## Discussion

In this study, we demonstrated the selective effects of metabolic activities on the segmentation clock processes. ETC and ATP synthesis inhibitions exclusively affected Hes7 intron processing and protein degradation processes, respectively, among the three key molecular processes, whereas glycolysis inhibition selectively affected Hes7 protein degradation and production processes (Fig. 6h). Although metabolism had been an attractive candidate for a global modulator of the segmentation clock tempo due to the universal role of energy, individual metabolic inhibitions were not as universal as temperature change which affected all three processes simultaneously. Instead, the combination of glycolysis and ETC inhibitions affected the three key molecular processes, synergistically extending the segmentation clock period.

How distinct metabolic inhibitions selectively affect the individual molecular processes remains elusive. As bio-energetic rates or the NAD+/NADH ratio did not directly correlate with the segmentation clock period or key kinetic parameters, specific metabolites and metabolic factors that are regulated by distinct metabolic inhibitors, such as glycolytic metabolites[36] and pentoses, may modulate the individual processes. It is also noteworthy that many metabolic enzymes and metabolites have moonlighting functions affecting cellular signaling pathways in addition to the canonical bio-energetic function[34,36,39–41], and that metabolic inhibitors can perturb other branches of metabolic pathways[42]. Clarifying the effectors of distinct metabolic inhibitions that modulate the three key molecular processes of the segmentation clock serves as a future research avenue. Importantly, even in the presence of multiple moonlighting functions and unexpected side effects in the metabolic pathways, all the metabolic inhibitions employed in this study exhibited selective effects.

As any single metabolic activities examined in this study are not a global modulator for the three key molecular processes of the segmentation clock, a question remains about how the kinetic parameters of these processes strongly scale with the clock period across multiple animal species (Fig. 1c). One possibility is that the three molecular processes may have been individually fine-tuned by combinations of separate mechanisms, including distinct metabolic activities, throughout evolution. As the segmentation clock operates during phylotypic stages where embryos of related species display the highest degree of morphological and molecular resemblance[43,44], the three kinetic parameters may be under evolutionary constraints. Different species may employ similar combinations of modulation mechanisms, considering that mouse and human PSM cells exhibited qualitatively similar responses to metabolic inhibitions. By contrast, considering that six animal species show apparently random glycolytic and mitochondrial respiration rates[27], these species may employ different combinations of modulation mechanisms. Even if there is no singular global modulator for the segmentation clock or developmental tempo, identifying selective modulation mechanisms and their combinations in individual species remains a valuable research avenue. Such

investigations will contribute to an evolutionary understanding of developmental tempo and then offer diverse strategies to manipulate it.

## Methods

### Stem cell cultures and PSM cell induction

Mouse EpiSCs (from RIKEN BRC #AES0204)[45] were maintained on fibronectin-coated dishes with the DMEM-F12 maintenance medium containing 15% Knockout Serum Replacement, Glutamax (2 mM), non-essential amino acids (0.1 mM), β-mercaptoethanol (0.1 mM), Activin A (20 ng/ml), bFGF (10 ng/ml), and IWR-1-endo (2.5 μM). Cells were passaged with ROCK inhibitor Y-27632 (10 μM). The medium was changed every day. For mouse PSM induction, $5 \times 10^4$ mouse EpiSCs were seeded on a Matrigel-coated 35 mm dish and cultured in the maintenance medium without IWR-1 for one day. Then the medium was changed into CDMi containing SB431542 (10 μM), CHIR99021 (10 μM), DMH1 (2 μM), and bFGF (20 ng/ml); this medium is hereafter referred to as the SCDF medium. The cells were cultured in the SCDF medium for 30 h.

Human iPSCs (201B7 line, from RIKEN BRC #HPS0063)[46] were cultured on Matrigel-coated dishes in the StemFit medium (Ajinomoto). Cells were passaged with ROCK inhibitor Y-27632 (10 μM). Ethical approval for human iPSC usage was granted by Department de Salut de la Generalitat de Catalunya (Carlos III Program). Our human PSM induction protocols have two variations: 1-step and 2-step protocols. For the 1-step protocol, $2 \times 10^4$ human iPSCs were seeded on a Matrigel-coated 35 mm dish and cultured for three days. Then the cells were cultured in the SCDF medium for three more days. For the 2-step protocol, $2 \times 10^4$ human iPSCs were seeded on a Matrigel-coated 35 mm dish and cultured for three to four days. Then the medium was changed into CDMi containing bFGF (20 ng/ml), CHIR99021 (10 μM), and Activin A (20 ng/ml), and the cells were cultured for one day. The cells were further cultured in the SCDF medium for one more day.

### DNA constructs and reporter cell lines

For the Hes7 oscillation assay, the m*Hes7* promoter-Firefly luciferase (FLuc)-NLS-PEST-stop-m*Hes7* (w/o intron) construct was used. For the Hes7 protein degradation assay, the reverse TetOne (rTetOne) promoter-m*Hes7* (w/o intron)-NLuc construct was used. For the Hes7 intron delay assay, the m*Hes7* promoter-NLuc-NLS-PEST-stop-m*Hes7* (w/o intron or w/ intron) construct was introduced into the cell line for the Hes7 oscillation assay. For the Hes7 production assay, the TetOne promoter-m*Hes7* (w/o intron)-NLuc construct was used. For the measurements in human cells, the constructs derived from h*HES7* sequences were used. All components are described previously[25]. All promoters or genes were subcloned into pDONR vector to create entry clones. These entry clones were recombined with a piggyBac vector[47] (a gift from K. Woltjen) or Tol2 vector[48] (a gift from K. Kawakami) by using the Multisite Gateway technology (Invitrogen). The constructs were stably introduced into the human iPSCs and mouse EpiSCs by electroporation with a 4D Nucleofector (Lonza) or using lipofectamine (Invitrogen).

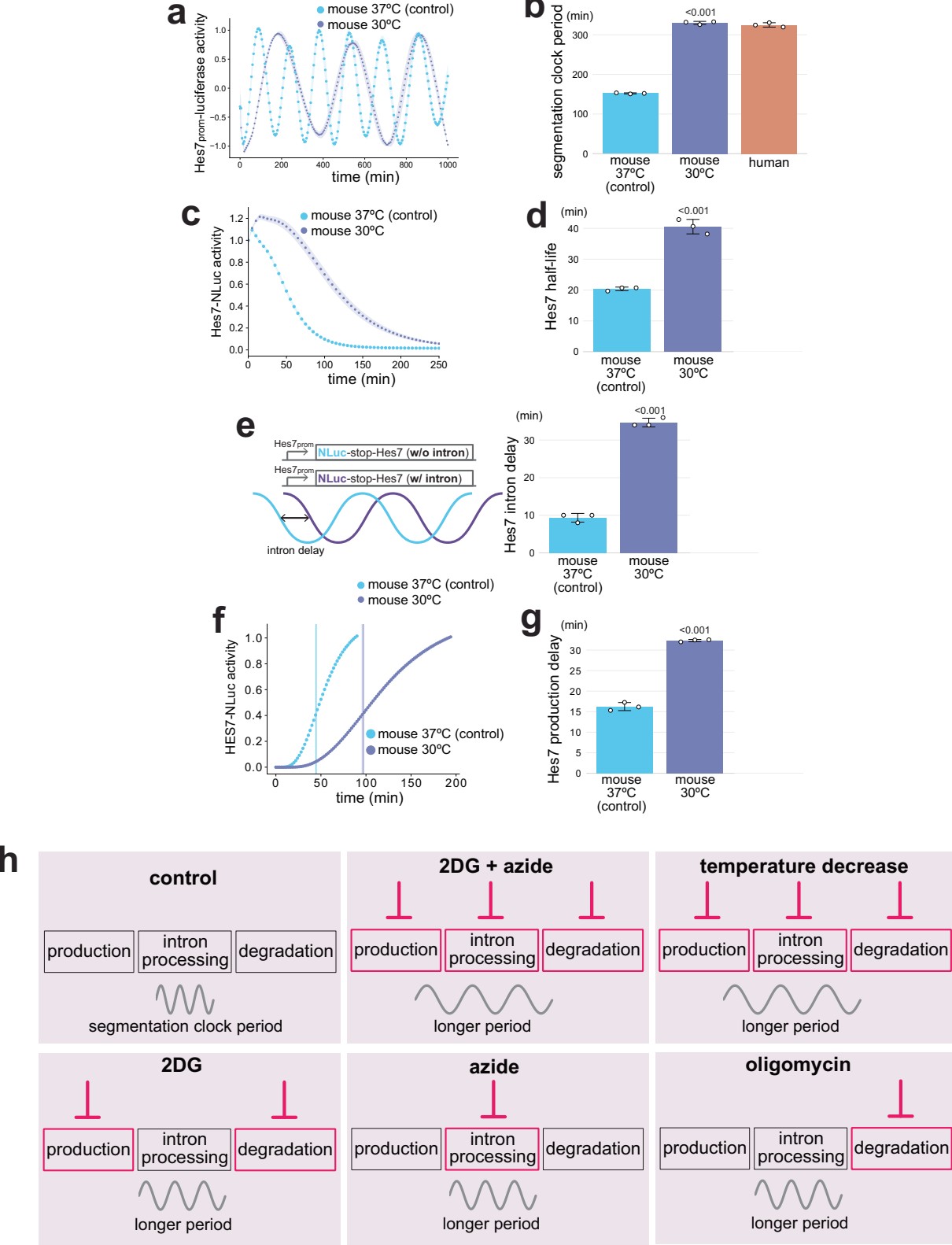

## Hes7 oscillation assay

As described previously[27], the mouse EpiSCs or human iPSCs stably expressing the oscillation reporter were used. After the PSM induction (the 1-step protocol was used for human iPSCs), the medium was changed into the SCDF medium containing a lower dosage of CHIR99021 (1 μM) and D-luciferin to monitor the oscillations of the *Hes7* promoter-luciferase reporter signal. The PSM cells were pretreated with metabolic inhibitors 0–4 h before the medium change, and kept in the medium containing each metabolic inhibitor during the measurement. The luminescence signal of the whole dish was measured at each time point with Kronos Dio Luminometer (Atto). The obtained entire traces were analyzed with pyBOAT 0.9.12[49]. A threshold of 500 (human and mouse 30 °C) or 300 (the rest) min was used for Sinc-detrending and amplitude normalization of the signal. The

**Fig. 6 | Global effects of temperature change on segmentation clock processes.**
**a** Effects of temperature decrease on the segmentation clock period. Mouse PSM
cells were incubated at 30 °C from minus 3 h, and the oscillatory activity of the *Hes7*
promoter-luciferase reporter was monitored. The signal was detrended and
amplitude-normalized. **b** Hes7 oscillation periods estimated from **a**. Human data
are from Fig. 5. **c** Effects of temperature decrease on Hes7 protein degradation.
Mouse PSM cells were incubated at 30 °C from minus 3 h, and the decay of the
Hes7-NLuc signal was monitored. **d** Hes7 half-lives estimated from **c** and Supple-
mentary Fig. 16b. **e** Effects of temperature decrease on Hes7 intron delay. Mouse
PSM cells were incubated at 30 °C from minus 3 h, and the oscillatory activities of
the luciferase reporters w/o and w/ *Hes7* intron sequences were monitored. The
intron delays were estimated from Supplementary Fig. 16c–e. **f** Effects of

temperature decrease on Hes7 production delay. Mouse PSM cells were incubated
at 30 °C from minus 3 h, and the onset of the Hes7-NLuc signal was monitored.
Vertical lines indicate the inflection points. **g** Hes7 production delays estimated
from **f** and Supplementary Fig. 16f. **a**–**g** Mouse 37 °C data are from Fig. 2.
**a**, **c**, **f** Shading indicates mean ± sd ($n = 3$). **b**, **d**, **e**, **g** Graphs indicate mean ± sd
($n = 3$). *P*-values are from two-sided Student's t-test. **h** Proposed scheme. While both
metabolic inhibitions and temperature decrease extend the segmentation clock
period, metabolic inhibitions selectively affect the three key molecular processes:
2DG extends Hes7 production delay and decelerates the protein degradation; azide
extends Hes7 intron delay; oligomycin decelerates Hes7 protein degradation. By
contrast, the combination of 2DG and azide affects all three processes like tem-
perature change. Source data are provided as a Source Data file.

processed signal was then analyzed using wavelets with periods of
200–500 (human and mouse 30 °C) or 100–300 (the rest) min. A
Fourier estimate of the wavelet analysis provided a distribution of
periods and their corresponding power. The period with the maximum
power for each of the signals was considered.

### Hes7 protein degradation assay
As described previously[27], the mouse EpiSCs and human iPSCs sta-
bly expressing Hes7 degradation reporters were used. PSM cells were
induced in the presence of Dox (100 ng/ml). The 2-step pro-
tocol was used for human iPSCs. The expression of the NLuc-fusion
proteins was initiated by washing out Dox and changing the med-
ium into CDMi containing protected furimazine (Promega). Meta-
bolic inhibitors were added concurrently with the Dox washout.
After the NLuc signal was confirmed 4–5 h later, the expression of
the fusion protein was halted by Dox (100 ng/ml) addition, and the
decay of NLuc signal was monitored with Kronos Dio luminometer.
To estimate Hes7 half-life, the slope of log2-transformed data was
calculated. A RANSAC algorithm (scikit-learn) was used to find the
most linear part of the decay curve and avoid the influence of
remaining mRNA.

### Hes7 intron delay assay
As described previously[27], the mouse EpiSCs and human iPSCs stably
expressing *Hes7* promoter-NLuc-stop-*Hes7* (w/ intron) and *Hes7* pro-
moter-FLuc-stop-*Hes7* (w/o intron) were used. After PSM cells were
induced (the 1-step protocol was used for human iPSCs), the medium
was changed into CDMi containing protected furimazine and D-luci-
ferin, and the oscillations of the NLuc and FLuc signals were simulta-
neously monitored with Kronos Dio luminometer. The PSM cells were
pretreated with metabolic inhibitors 0-4 h before the medium change,
and kept in the medium containing each metabolic inhibitor during the
measurement. To estimate Hes7 intron delay, first, the oscillatory
traces were detrended and amplitude normalized using pyBOAT
0.9.12[49] as described previously. Then, the oscillation phase difference
between the reporter w/o intron and the one w/ intron was estimated
by calculating their cross-correlation with Python (SciPy). To normal-
ize the difference in the maturation/degradation time between NLuc
and FLuc, cells containing the *Hes7* promoter-NLuc-stop-*Hes7* (w/o
intron) and *Hes7* promoter-FLuc-stop-*Hes7* (w/o intron) constructs
were also used, and the phase difference between the NLuc and FLuc
reporters was subtracted from that between the w/o intron and w/
intron reporters.

### Hes7 production delay assay
As described previously[25], the mouse EpiSCs and human iPSCs stably
expressing the production delay assay reporters were used. After PSM
cells were induced in the absence of Dox, the medium was changed
into CDMi containing protected furimazine and metabolic inhibitors.
Four hours after the medium change, the expression of the fusion
protein was initiated by Dox (100 ng/ml), and the onset of NLuc signal
was monitored with Kronos Dio luminometer. To estimate Hes7

production delay, the following model was considered.

$$\frac{dm}{dt} = -\delta_m m \quad (t < \tau_{Tx}) \tag{1}$$

$$\frac{dm}{dt} = \beta_T - \delta_m m \quad (t \geq \tau_{Tx}) \tag{2}$$

$$\frac{dp}{dt} = \alpha m(t - \tau_{Tl}) - \delta_p p \tag{3}$$

where *m* and *p* are the concentrations of Hes7 mRNA and protein, $\delta_m$
and $\delta_p$ are the degradation rates of mRNA and protein, $\beta_T$ is the
transcription rate of the TetOne promoter, $\alpha$ is the translation rate, and
$\tau_{Tx}$ and $\tau_{Tl}$ are the transcription and translation delays.
  The solution of this is

$$p(t) = 0 \quad (t < \tau) \tag{4}$$

$$p(t) = \frac{a}{\delta_m - \delta_p}\left(e^{-\delta_m(t-\tau)} - \frac{\delta_m}{\delta_p}e^{-\delta_p(t-\tau)}\right) + \frac{a}{\delta_p} \quad (t \geq \tau) \tag{5}$$

where $\tau = \tau_{Tx} + \tau_{Tl}$, and $a = \alpha\beta_T/\delta_m$.
  $\delta_p$ was estimated in the degradation assay. The production delay
$\tau$, together with $\delta_m$ and a, was estimated by fitting the data from the
production delay assay to Eqs. (4), (5) with Python (SciPy)'s basin-
hopping algorithm. Data points within 2×(duration to reach the
inflection point) were used for fitting, and the inflection point was
determined by calculating the 2nd derivatives of the data.

### Oscillation period simulation
To simulate the Hes7 oscillation, previously described delay differ-
ential equations were used[25,27,29].

$$\frac{dm}{dt} = \frac{\beta}{1 + \left(\frac{p(t-\tau_m)}{K}\right)^n} - \delta_m m \tag{6}$$

$$\frac{dp}{dt} = \alpha m(t - \tau_p) - \delta_p p \tag{7}$$

where $\tau_m = \tau_{Rp} + \tau_{Tx} + \tau_{In}$, $\tau_p = \tau_{Tl}$.
  $\delta_p$ was estimated from the protein degradation assay, $\tau_{In}$ was
estimated from the intron delay assay, and $\tau_{Tx} + \tau_{Tl}$ and $\delta_m$ were
simultaneously estimated from the production delay assay. The other
parameters ($\alpha$, $\beta$, K, n, $\tau_{Rp}$) are the same as the values described
previously[25,27]. All parameter values are summarized in the Source Data
file. The period was estimated by calculating the peak-to-peak dis-
tance. Numerical calculations and period estimation were performed
with Python.

## Seahorse metabolic rate measurement

As described previously[27], PSM cells were re-seeded on a fibronectin-coated Seahorse plate (Agilent) at a density of $9.1 \times 10^5$ cells/cm$^2$ (mouse assay) or $7.3 \times 10^5$ cells/cm$^2$ (human assay) in 100 µl of Seahorse XF DMEM (Agilent) supplemented with glucose (20 mM), pyruvate (1 mM), and glutamine (2 mM). Note that this medium composition differs from the one used for the other kinetic parameter measurements. Cells were allowed to attach at RT for 10 min and then transferred to a 37 °C incubator without $CO_2$ for 10 min. After that time, 400 µl of Seahorse XF DMEM medium at 37 °C was added carefully to each well without disturbing the attached cells for a total of 500 µl. Cells were incubated at 37 °C without $CO_2$ for 30 more min. For the real-time ATP rate assay (Agilent), oligomycin (1 µM), rotenone (0.5 µM), and antimycin A (0.5 µM) were used. MitoATP could not be calculated as a parameter to compare across conditions since azide was added at the beginning in some conditions. The Wave Desktop and online app provided by the manufacturer were used for analysis.

## NAD + /NADH ratio measurement

After PSM induction, the medium was replaced with SCDF medium containing a low dose (1 µM) of CHIR99021 and simultaneously treated with metabolic inhibitors. After 4 h treatment, the cells were washed with PBS and lysed directly on the plate with 0.1 N NaOH with 0.5% dodecyltrimethylammonium bromide (DTAB). The lysate was measured for the NAD + /NADH ratio using the NAD/NADH-Glo™ Assay (promega) according to its protocol. The lysate was split into two portions: one treated with 0.4 N HCl and one untreated. Both portions were then heated at 60 °C for 10 min. The treated one was neutralized with Trizma® base, and HCL/Trizma® was added to the other to match the concentration. 50 µl of each was mixed with an equal volume of the reaction solution and incubated at room temperature for 30 min, after which the luminescence was measured.

## Imaging

After PSM induction, the medium was changed into the SCDF medium containing a lower dosage of CHIR99021 (1 µM). The metabolic inhibitors were added 4 h before the medium change, and the start of inhibitor treatment was defined as time 0. At each time point, cells were fixed in 4% formaldehyde, and bright-field images were acquired using Axiovert 5 (ZEISS).

## Sample definition

All sample replicates are from independent experiments (biological replicates).

## Reporting summary

Further information on research design is available in the Nature Portfolio Reporting Summary linked to this article.

## Data availability

Source data are provided with this paper.

## Code availability

The custom scripts used are available at Github [https://github.com/mebisuya/MetabolicActivities].

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

## Acknowledgements

We are grateful to Kristina Stapornwongkul, Hidenobu Miyazawa, and Shuting Xu for their comments on the manuscript. This work was supported by EMBL; the Deutsche Forschungsgemeinschaft (DFG, German Research Foundation) under Germany's Excellence Strategy - EXC 2068 – 390729961 - Cluster of Excellence Physics of Life of TU Dresden; the European Research Council (ERC) under the European Union's Horizon 2020 research and innovation program (grant agreement No. 101002564) (to M.E.); PRESTO (grant number JP20332265) from Japan Science and Technology Agency (JST) (to M.M.); the Boehringer Ingelheim Fonds (BIF) PhD fellowship (to J.L.); M.E. is supported by the Alexander von Humboldt Foundation in the framework of the Alexander von Humboldt Professorship endowed by the Federal Ministry of Education and Research. The human iPSC and mouse EpiSC lines were provided by the RIKEN BRC through the National BioResource Project of the MEXT, Japan.

## Author contributions

M.M. and M.E. designed the work and wrote the manuscript. M.M. performed the experiments and analyzed the data. J.L. analyzed the data. All authors contributed to the manuscript and approved the final version.

## Funding

## Competing interests

The authors declare no competing interests.
