## [Peer Review file · Nature Communications]

Metabolic activities are selective modulators for individual segmentation clock processes

Corresponding Author: Professor Miki Ebisuya

Version 0:

Reviewer comments:

Reviewer #1

(Remarks to the Author)

The manuscript "Metabolic activities are selective modulators for individual segmentation clock processes" by Matsuda et al. addresses the question of how developmental time is regulated across species. To address this, they test the hypothesis that metabolism affects timing of gene expression oscillations. They employ in vitro differentiation models of somitogenesis to compare the timing of Hes7 oscillations between mouse and human cells. They find that inhibition of glycolysis has different effects on the different steps of Hes7 oscillations (intron delay, protein production and degradation) from inhibition of the electron transport chain (ETC) or ATP synthase. Interestingly, they describe a synergistic effect of simultaneously inhibiting glycolysis and ETC. This study builds on a previous publication (Dias-Cuadros et al. 2023) that stated that the mass specific metabolic rate differs between mouse and human, in particular NAD⁺/NADH levels, which leads to differences in segmentation clock period. This is a well written manuscript and an interesting, detailed analysis of the effect of metabolic perturbations on the different steps of Hes7 oscillations and contributes to the understanding of how metabolism affects signalling oscillations.

There are multiple points that should be addressed to clarify findings of the current paper and put it into context of our current knowledge:

- In most cases, metabolic perturbations lead to dampened oscillations with near-absence of oscillations towards the end of imaging. In addition, in 10 mM 2DG, 10 nM Oligomycin and in all double-perturbations, the Hes7 signal is nearly absent (Figure S4), which for the double-perturbations is also mentioned in the text. Detrending and amplitude-normalization then masks these effects on overall signalling activity. To clarify the effect of metabolic perturbations on Hes7 dynamics, the following points should be addressed:

- o Raw data should be provided for all experimental perturbations, including those for quantification of intron delay and for human cells. In addition, it would be useful to show single tracks rather than signals averaged over multiple experiments to get a clear understanding of oscillation dynamics.
- o Signals barely above 0 raise the question whether this should be considered as oscillations at all. To clarify this, one question is how big the variation between samples is. These aspects should be discussed further.
- o The progressing dampening could be due to an overall progressive ceasing of oscillations in all cells, loss of oscillations in a subset of cells within the culture or desynchronization. It would be useful to provide representative movies of the samples, which could give indications on what is going on. I do however acknowledge that luminescence imaging does not allow quantification of single cells.
- o The strong effect of metabolic perturbation on Hes7 signal raises the question of how healthy the cell culture is. Via the pentose phosphate pathway, glycolysis inhibition has an effect on all anabolic processes in cells, including production of amino acids, nucleotides and fatty acids. Especially the simultaneous inhibition of glycolysis and ETC/ATP synthase could lead to senescence/ death of cells. The effect on the segmentation clock might therefore be more indirect. To rule this out, cell cultures should be stained for apoptosis markers at the end of the experiments and compared to control. In addition, brightfield movies could give indications of the state of cells. At the moment, there is no image or movies of cells shown in the manuscript.

In addition, quantification of ATP levels in cells upon double-block would be useful.

o Using metabolic perturbations, the authors show that it is possible to slow down the clock. To rule out the effect on cell viability, it would be more convincing to provide a condition, in which metabolic perturbations speed up the clock.

o A more balanced discussion on the observations would help.

- One question remains until the end of the paper: How do these parameters (including glycolysis and ETC) scale in different species. It has previously been suggested that different cell sizes lead to different mass specific metabolic rates in a species specific manner (Dias-Cuadros et al. 2023). This way, a global change in metabolism (including glycolysis and ETC) is expected between mouse and human. The authors should include this suggestion in their discussion. In this context, it would also be interesting to quantify the NAD⁺/NADH levels.

- The authors finally present temperature as global modulator of segmentation clock rates. This is a nice visualization of the Arrhenius equation, but quite expected.

- Some findings directly contradict a previous publication (Dias-Cuadros et al. 2023). For instance, they had found that ETC inhibition interferes with translation by analysing global effects on translation. This should be addressed and discussed in the paper. What is different about this Hes7 reporter construct?

- The authors show that blocking multiple metabolic pathways has a synergistic effect on the segmentation clock. At the same time, they conclude that metabolism is not a global modulator, because each metabolic pathway on its own has not a global effect. This however does not exclude metabolism as a whole as modulator.

- Here, the authors focus on the analysis of how metabolism affects oscillations of one protein. Even though this is a core component of the clock, a global mechanism should have affect expression and protein dynamics in general. Experimental details or at least a detailed discussion would put their work into perspective.

Minor points:

- The authors mention to study a "cell-autonomous" (p. 3, l. 9). As they are not analysing cellular effects but rather cell populations, this should be rephrased.

- It has already been found in Dias-Cuadros 2023 that energy is not the main reason for an effect on the segmentation clock. This should be acknowledged (p. 7, line 2).

- Some of the selective effects of metabolism on the different steps of gene expression oscillations seem to be expected. The effect of glycolysis inhibition on protein production seems to be obvious based on the fact that glycolysis feeds into the pentose phosphate pathway which is required for various anabolic processes. It would be interesting to discuss such potential links in more detail.

- The Methods section could sometimes be more extensive, for instance on metabolic perturbations or details when is the period quantified (the whole track or only the end?).

Reviewer #2

(Remarks to the Author)

Developmental rates are different across species, and we know little as to what drives the differences. The group in the past had shown that the speed of biochemical processes for the gene Hes7 are associated to the speed of Hes7 oscillations. It has been shown that cytosolic NAD⁺/NADH ratio is upstream of protein production in presomitic mesoderm cells and controls the period of Hes7 oscillations.

In this new work, Matsuda et al. dissect the contribution of specific well-established metabolic inhibitors to the period of Hes7. They exploit their ability to individually measure the speed of specific biochemical processes (production delay, intron delay and protein degradation) for the period of Hes7 as a readout of the perturbations.

They find that inhibition of glycolysis affects predominantly the production and degradation of Hes7 in mouse and human cells whereas blocking the ETC transport chain impacts Hes7 intron delay.

This work is important for the field as it dissects specific effects of metabolic components. It demonstrates that (1) inhibition of metabolites causes selective responses in Hes7 kinetic parameters and (2) individual metabolic perturbations do not recapitulate a global shifting of rate like changes of temperature do for Hes7 oscillations. The main criticism would be that the authors do not contextualize their work in terms of the NAD⁺/NADH ratio in the cytoplasm.

Comments:

1. The authors dissect Hes7 parameters and evaluate OCR and glycoATP as a proxy for energetic use in cells. As it has been previously proposed that the redox state (cytosolic NAD⁺/NADH ratio) is more important than ATP for developmental rate in PSM cells between mouse (Diaz-Cuadros et al., 2023), I wonder if this work would be more pertinent/appropriate if they included measurements of the redox state (cytosolic NAD⁺/NADH ratio) as well:

> At minimum, they could show NAD⁺/NADH ratio in their 2DG perturbation in mouse. Ideally, it would be good to see the NAD⁺/NADH ratio in 1mM 2DG + 1mM azide perturbation as well since the combination shows a global perturbation of Hes7 period similar to changes in temperature. Moreover, Supp Fig 5 indicates a PER comparable between 2DG+azide treatment and controls. Instead, 2DG only treatment reduces PER. This may imply recovery of glycolytic function in the 2DG+azide and suggests that when both pathways are inhibited the cells adjust to increase PER/redox. This may have implications in terms of global effects of energy consumption/redox in the period of Hes7: this treatment rewires the energetic use of cells potentially towards glycolysis which the authors should discuss. Have they considered consecutive addition of 2DG followed by azide treatments instead of combinatorial?

> Given that the NAD⁺/NADH ratio parameter has been found to be upstream of Hes7 oscillations, it would be good to understand how they think its perturbation affects Hes7 production delay and splicing. According to Diaz-Cuadros et al., we would not expect to see an effect on Hes7 degradation.

2. The titration and characterization for 2DG, azide and oligomycin treatments is very neat. Oscillations in 2DG treatment or azide+2DG are dampened and quite hard to see in comparison to the controls. What quality controls were performed to determine the viability of the cells before/after treatment? It would be important to understand if control and treated cells are both as viable for the comparisons.

3. All treatments to measure Hes7 parameters and the Seahorse measurements are done in medium supplemented with glucose, pyruvate and glutamine. Are the concentrations in the medium and Seahorse medium for these parameters the same? If they aren't, I wonder to what extent the responses measured for Hes7 period parameters and the Seahorse are comparable. Would we expect them to be affected by the default concentrations?

4. The authors conclude that metabolic activities may not act as global modulators for the segmentation clock tempo. However, the combinatorial addition of glycolytic and ETC inhibitors (1mM 2DG + 1mM azide and/or oligomycin) in mouse PSM cells affects the three parameters of Hes7. To me, this result somewhat opposes their interpretation as I can imagine a scenario where the combined treatment with inhibitors mimics a global endogenous regulator (enzyme/metabolite) that rewires energy consumption in cells to increase solely PER. This perturbation is to me the most interesting/puzzling metabolic effect. It is also a complicated experiment to put a lot of emphasis on because the oscillations are quite dampened, but I think that some additional info and tests would strengthen the findings:

> As mentioned earlier, they observe that the inhibition reduces OCR but not glycoATP, and state that they were expecting a synergistic effect on energy consumption. How do the authors think that the cells are buffering the response to 2DG when they add azide? Do they observe a similar energetic phenotype on the Seahorse if they treat cells with 2DG and Oligomycin?

> How do they interpret the difference in the results obtained in 2DG+Oligomycin versus 2DG+Azide for intron delay and production delay?

> How would these perturbations look in human?

> Given that the perturbation seems to have a global effect for Hes7, do they observe a change in the overall speed of differentiation beyond Hes7?

5. Production delay serves as a proxy for "a combined delay caused by the gene expression steps of Hes7, including transcription and translation, except for intron-related steps". I wonder if the authors could unpick whether the perturbations they measure affect more transcription or translation?

6. The authors show that perturbation temperature affects Hes7 parameters and conclude that temperature acts as a global regulator for tempo. While cells from various species grown at similar temperatures in vitro show differences in Hes7 period, this perturbation may be insightful to understand changes in the overall rate of tempo. At present the association between the period of Hes7 and the rate of development is still correlative. What other phenotypic effects do the authors observe in cells? And at higher temperatures? It would be good to verify that shifts in temperature indeed affect changes in tempo globally and not just to Hes7.

Reviewer #3

(Remarks to the Author)

Matsuda et al. investigate how specific metabolic interventions affect the kinetics of the key molecular processes that determine the segmentation clock period. They employ previously established assays to measure the duration of protein half-lives, protein production and intron delay of the Hes7 gene product together with the segmentation clock period upon perturbations of glycolysis, the electron transport chain and ATP production. All perturbations extended the clock period but affected different molecular processes underlying the clock period. Although the mechanistic basis of the linkage between individual perturbations and the kinetics of specific molecular processes has not been investigated, this is a new and unexpected finding and an important contribution to the field of developmental timing as it shows how timing can be modulated through different metabolic and molecular processes. Finally, the authors show that combined perturbations exhibited partially synergistic effects, that the selective effects of metabolic perturbations are shared between mouse and human, and that the effect of temperature on clock period emerges from changes in the kinetics of all underlying molecular processes.

The manuscript has a clear structure and is easy to follow, so are the figures.

Specific points

Major

1. One major concern with the manuscript in its current form is that the authors do not show any primary data of the cells under the different perturbations. One would expect that – depending on the concentrations used - the inhibitors have generic effects on cell viability. For the reader to evaluate how well cells tolerate the perturbations, the authors should show cell images or movies of at least a few representative cases, e.g. for the experiment shown in Supplementary Figure 4 where inhibitor treatments lowered the Hes7 reporter signal drastically. How well can peaks be identified and how long do cells keep oscillating under these conditions? It would also be interesting to know how cell proliferation is affected by the different perturbations.

2. Figure 4b: Can the authors explain why 1 mM 2DG and 1 mM azide separately do not have an effect on protein half-life but together they increase half-life? Figure 2e shows that 10 mM 2DG increases half-life to a similar degree as 1 mM 2DG and 1 mM azide combined. Could the combined effect result from a generally lower metabolism compared to the single inhibitor treatment that is also seen when increasing the 2DG dose to 10 mM?

3. Are the changes to the kinetics of specific molecular processes sufficient to explain the changes in the clock period? In a previous publication (PMID 32943519), the authors used a mathematical model to study the dependency of the segmentation clock period on the kinetics of different biochemical reactions. Using this model to test how well the inhibitor treatments explain changes in the clock period would give the manuscript a significant lift.

Minor

4. Regarding the intron delay assay, please explain the constructs in more detail, e.g. in the methods section or figure legend. Do the constructs contain any Hes7 coding sequence and could therefore feed back on the clock oscillations? What is the length, identity and position of the introns in the intron-containing construct?

5. In Supplementary Figure 1 (generally in the supplementary figures), do the three plots per condition represent replicates? If so, please state in figure legend.

Reviewer #4

(Remarks to the Author)

Version 1:

Reviewer comments:

Reviewer #1

(Remarks to the Author)

The authors have addressed my comments sufficiently, have added further control experiments and single-cell traces and clarifications in the text.

Reviewer #2

(Remarks to the Author)

Reviewers have addressed all concerns and contextualize their work in terms of the NAD⁺/NADH ratio in the cytoplasm.

Reviewer #3

(Remarks to the Author)

In their revised manuscript, Matsuda et al. have performed several new experiments, provided additional primary data, and made several small additions to the text to address all of the reviewers' concerns. The main concern shared by all three reviewers was that the metabolic inhibitions affect cell viability, raising questions how specific the observed changes to segmentation clock were. The new data to address these concerns show that cells remain viable and continue to oscillate (albeit with low amplitude) for the first 10h after the addition of inhibitors, but that there is massive cell death thereafter. While these findings leave open the possibility that what has been analyzed are not steady-state oscillations but rather a transient behaviour towards cell death, there is now enough data in the manuscript for each reader to evaluate these possibilities for themselves.

As suggested by this reviewer, the authors have performed simulations to compare the observed effects of metabolic interventions on clock period with theoretical predictions. They find a qualitative agreement, although the quantitative differences (the observed period lengthening seems to be consistently larger than the predicted one) could have been acknowledged more explicitly.

With respect to mechanistic relationships between metabolic perturbations, Hes7 kinetic parameters and clock period, there remain some discrepancies with previous work in the field (PMID: 36599986), but this is to be expected and will stimulate further work and progress in the field.

In reply to Reviewer 1:

The manuscript “Metabolic activities are selective modulators for individual segmentation clock processes” by Matsuda et al. addresses the question of how developmental time is regulated across species. To address this, they test the hypothesis that metabolism affects timing of gene expression oscillations. They employ in vitro differentiation models of somitogenesis to compare the timing of Hes7 oscillations between mouse and human cells. They find that inhibition of glycolysis has different effects on the different steps of Hes7 oscillations (intron delay, protein production and degradation) from inhibition of the electron transport chain (ETC) or ATP synthase. Interestingly, they describe a synergistic effect of simultaneously inhibiting glycolysis and ETC. This study builds on a previous publication (Dias-Cuadros et al. 2023) that stated that the mass specific metabolic rate differs between mouse and human, in particular NAD⁺/NADH levels, which leads to differences in segmentation clock period. This is a well written manuscript and an interesting, detailed analysis of the effect of metabolic perturbations on the different steps of Hes7 oscillations and contributes to the understanding of how metabolism affects signalling oscillations.

There are multiple points that should be addressed to clarify findings of the current paper and put it into context of our current knowledge:

- In most cases, metabolic perturbations lead to dampened oscillations with near-absence of oscillations towards the end of imaging. In addition, in 10 mM 2DG, 10 nM Oligomycin and in all double-perturbations, the Hes7 signal is nearly absent (Figure S4), which for the double-perturbations is also mentioned in the text. Detrending and amplitude-normalization then masks these effects on overall signalling activity. To clarify the effect of metabolic perturbations on Hes7 dynamics, the following points should be addressed:

1-1. We appreciate the reviewer’s comments. It was not our intention to imply that metabolic inhibitions can slow down segmentation clock processes without any adverse effects, and we completely agree to describe those ‘side effects’ in more detail. Our specific responses are as follows:

o Raw data should be provided for all experimental perturbations, including those for quantification of intron delay and for human cells. In addition, it would be useful to show single tracks rather than signals averaged over multiple experiments to get a clear understanding of oscillation dynamics.

1-2. The single tracks of raw oscillation data were added to all relevant figures, including those for quantification of the intron delay and human cells (new Supplementary Figs. 1b; 4b,c; 6a-c; 9b,c; 11a,b, 13b,c; 16a,d,e).

o Signals barely above 0 raise the question whether this should be considered as oscillations at all. To clarify this, one question is how big the variation between samples is. These aspects should be discussed further.

1-3. We agree that the samples treated with metabolic inhibitors, especially 2DG and oligomycin, show low signals of the luciferase reporter. Even in those samples, however, the oscillations are relatively clear (see single tracks in new Supplementary Figs. 1b; 6a-c; 11a,b). These oscillations were less visible in the averaged signals of the original figures (now new Supplementary Figs. 1a; 6a-c; 11a,b) mostly because the Y-axis was adjusted to match the high signals of control samples.

o The progressing dampening could be due to an overall progressive ceasing of oscillations in all cells, loss of oscillations in a subset of cells within the culture or desynchronization. It would be useful to provide representative movies of the samples, which could give indications on what is going on. I do however acknowledge that luminescence imaging does not allow quantification of single cells.

1-4. As explained in point 1-3, oscillations are relatively clear even in samples with lower reporter signals. In some samples, however, the oscillations indeed begin to dampen after ~10 hours. This might be because the cells start dying after a long exposure to metabolic inhibitors (see point 1-5). Unfortunately, assessing potential desynchronization is technically challenging with our current luminescence imaging setup.

o The strong effect of metabolic perturbation on Hes7 signal raises the question of how healthy the cell culture is. Via the pentose phosphate pathway, glycolysis inhibition has an effect on all anabolic processes in cells, including production of amino acids, nucleotides and fatty acids. Especially the simultaneous inhibition of glycolysis and ETC/ ATP synthase could lead to senescence/ death of cells. The effect on the segmentation clock might therefore be more indirect. To rule this out, cell cultures should be stained for apoptosis markers at the end of the experiments and compared to control. In addition, brightfield movies could give indications of the state of cells. At the moment, there is no image or movies of cells shown in the manuscript.

In addition, quantification of ATP levels in cells upon double-block would be useful.

1-5. This is an important point. To assess the state of cells, we took bright-field images at 0, 7, and 21 hours after several metabolic inhibitor treatments (new Supplementary Fig. 1c). Cells seemed relatively healthy at 7 hours. At 21 hours, however, cells under harsh metabolic inhibitions, such as 2DG+azide and 2DG+oligomycin, showed much lower densities and more floating cells than the control, suggesting significant cell death.

Based on this data, we concluded that measurements within 10 hours are not severely impacted by cell death. The degradation rate and production delay assays are quick enough. In contrast, we should be careful about the oscillation assays for the period and intron delay quantifications. Luckily, even though the oscillation sometimes began to dampen after 10 hours, the periods per se were relatively consistent across the entire time course.

The ATP production rate (OCR and glycoATP) under the 2DG+azide block was measured in old Fig. 3g (new Fig. 3g).

o Using metabolic perturbations, the authors show that it is possible to slow down the clock. To rule out the effect on cell viability, it would be more convincing to provide a condition, in which metabolic perturbations speed up the clock.

1-6. While we understand the reviewer's point, accelerating the segmentation clock is more challenging than decelerating it. We are unaware of any pharmacological metabolic inhibitors or activators capable of achieving significant acceleration. For example, although 2DG treatment increases the NAD⁺/NADH ratio (see point 1-8) that was reported to regulate the segmentation clock (PMID: 36599986), 2DG did not accelerate but decelerate the segmentation clock.

Even though one genetic perturbation has been reported to accelerate the clock (PMID: 36599986), the degree of acceleration observed is modest, and it will be difficult to further decompose the small difference into three segmentation clock processes.

o A more balanced discussion on the observations would help.

1-7. Following the suggestion, we stated: "It must be noted, however, that metabolic inhibitions, especially 2DG and oligomycin treatment, lowered the reporter signal (Supplementary Fig. 1a; 6) and that prolonged exposure to combined metabolic inhibitions often resulted in cell death (Supplementary Fig. 1c)".

- One question remains until the end of the paper: How do these parameters (including glycolysis and ETC) scale in different species. It has previously been suggested that different cell sizes lead to different mass specific metabolic rates in a species specific manner (Dias-Cuadros et al. 2023). This way, a global change in metabolism (including glycolysis and ETC) is expected between mouse and human. The authors should include this suggestion in their discussion.

In this context, it would also be interesting to quantify the NAD⁺/NADH levels.

1-8. Following the suggestion, we measured the NAD⁺/NADH redox balance in mouse PSM cells treated with several metabolic inhibitors (new Fig. 3h). Glycolysis inhibitor 2DG increased the NAD⁺/NADH ratio while ETC inhibitor azide decreased it, as expected. However, there was no correlation between the NAD⁺/NADH ratio and the segmentation clock period across five metabolic inhibition conditions. Thus, we cannot say that the NAD⁺/NADH is the single important metabolic factor dictating the clock period.

Regarding the mass-specific metabolic rates, we previously measured the PSM cell volumes and metabolic rates (the glycolysis rate and OCR) of six mammalian species (PMID: 37343565). We could not find any correlation between volume-specific metabolic rates and the segmentation clock period. This was discussed in the manuscript: “considering that six animal species show apparently random glycolytic and mitochondrial respiration rates, these species may employ different combinations of modulation mechanisms.”

- The authors finally present temperature as global modulator of segmentation clock rates. This is a nice visualization of the Arrhenius equation, but quite expected.

1-9. We agree with the reviewer that this result is expected. Temperature shift was used as a ‘positive control’ that contrasts with metabolic inhibitions rather than a surprising discovery. We stated, “To contrast the selective modulation by metabolic inhibitions, we employed a temperature shift, a well-established method to modulate the segmentation clock in ectotherms”.

- Some findings directly contradict a previous publication (Dias-Cuadros et al. 2023). For instance, they had found that ETC inhibition interferes with translation by analysing global effects on translation. This should be addressed and discussed in the paper. What is different about this Hes7 reporter construct?

1-10. Diaz-Cuadros et al. 2023 demonstrated the impact of ETC inhibition on bulk translation, but they did not examine its effect on Hes7 translation. In contrast, we focused on the kinetic parameters of Hes7. In addition, the translation ‘rate’ measured in Diaz-Cuadros et al and the production ‘delay’ measured by us are different concepts. The rate represents production per time, while the delay represents the onset timing of production.

- The authors show that blocking multiple metabolic pathways has a synergistic effect on the segmentation clock. At the same time, they conclude that metabolism is not a global modulator, because each metabolic pathway on its own has not a global effect. This however does not exclude metabolism as a whole as modulator.

1-11. We are sorry that our intention was not clear enough. We meant that ‘individual’ metabolic activities are selective while ‘combinations’ of multiple metabolic activities can be global.

We slightly modified the wording to highlight the difference between individual and combined metabolic activities, discussing this topic in the following paragraphs:

“It is noteworthy that the combination of glycolysis and ETC inhibitions affected the three key molecular processes simultaneously (1 mM 2DG + 1 mM azide in Fig. 4), fulfilling the criteria as a global modulator for the segmentation clock tempo.”

“Although metabolism had been an attractive candidate for a global modulator of the segmentation clock tempo due to the universal role of energy, individual metabolic inhibitions were not as universal as temperature change which affected all three processes simultaneously. Instead, the combination of glycolysis and ETC inhibitions affected the three key molecular processes, synergistically extending the segmentation clock period.”

“As any single metabolic activities examined in this study are not a global modulator for the three key molecular processes of the segmentation clock”.

- Here, the authors focus on the analysis of how metabolism affects oscillations of one protein. Even though this is a core component of the clock, a global mechanism should have affect expression and protein dynamics in general. Experimental details or at least a detailed discussion would put their work into perspective.

1-12. As noted by the reviewer, our study focuses on determining whether metabolic activities act as a global modulator of the three key molecular processes of Hes7. While examining effects on other genes falls outside the scope of this work, we recently demonstrated the impact of the glycolysis inhibitor 2DG on broader protein degradation rates using quantitative mass spectrometry (Matsuda et al, bioRxiv 2024, <https://doi.org/10.1101/2024.06.07.597977>). This led us to state, “We recently observed this decelerating effect of 2DG on the degradation of numerous other proteins as a consistent trend”.

Minor points:

- The authors mention to study a “cell-autonomous” (p. 3, l. 9). As they are not analysing cellular effects but rather cell populations, this should be rephrased.

1-13. Our intention was solely to distinguish the mechanisms of oscillation and synchronization. We have deleted the sentence: “Even though the individual oscillations are further synchronized among cells, this study focuses on the cell-autonomous oscillation mechanism”.

- It has already been found in Dias-Cuadros 2023 that energy is not the main reason for an effect on the segmentation clock. This should be acknowledged (p. 7, line 2).

1-14. We have modified the sentence and cited Diaz-Cuadros et al paper: “These results confirmed the efficacy of the metabolic inhibitors and suggested that the segmentation clock period did not directly correlate with the bio-energetic rates, at least with the OCR or glycoATP, consistent with the previous report”.

- Some of the selective effects of metabolism on the different steps of gene expression oscillations seem to be expected. The effect of glycolysis inhibition on protein production seems to be obvious based on the fact that glycolysis feeds into the pentose phosphate pathway which is required for various anabolic processes. It would be interesting to discuss such potential links in more detail.

1-15. Following the suggestion, we discussed: “specific metabolites and metabolic factors that are regulated by distinct metabolic inhibitors, such as glycolytic metabolites and pentoses, may modulate the individual processes”.

- The Methods section could sometimes be more extensive, for instance on metabolic perturbations or details when is the period quantified (the whole track or only the end?).

1-16. We modified the method to include more details. The whole trace was used for the period quantification, and the timings of metabolic perturbations are also described in the figure legends.

In reply to Reviewer 2:

Developmental rates are different across species, and we know little as to what drives the differences. The group in the past had shown that the speed of biochemical processes for the gene Hes7 are associated to the speed of Hes7 oscillations. It has been shown that cytosolic NAD⁺/NADH ratio is upstream of protein production in presomitic mesoderm cells and controls the period of Hes7 oscillations.

In this new work, Matsuda et al. dissect the contribution of specific well-established metabolic inhibitors to the period of Hes7. They exploit their ability to individually measure the speed of specific biochemical processes (production delay, intron delay and protein degradation) for the period of Hes7 as a readout of the perturbations.

They find that inhibition of glycolysis affects predominantly the production and degradation of Hes7 in mouse and human cells whereas blocking the ETC transport chain impacts Hes7 intron delay.

This work is important for the field as it dissects specific effects of metabolic components. It demonstrates that (1) inhibition of metabolites causes selective responses in Hes7 kinetic parameters and (2) individual metabolic perturbations do not recapitulate a global shifting of rate like changes of temperature do for Hes7 oscillations. The main criticism would be that the authors do not contextualize their work in terms of the NAD⁺/NADH ratio in the cytoplasm.

Comments:

1. The authors dissect Hes7 parameters and evaluate OCR and glycoATP as a proxy for energetic use in cells. As it has been previously proposed that the redox state (cytosolic NAD⁺/NADH ratio) is more important than ATP for developmental rate in PSM cells between mouse (Diaz-Cuadros et al., 2023), I wonder if this work would be more pertinent/appropriate if they included measurements of the redox state (cytosolic NAD⁺/NADH ratio) as well:

> At minimum, they could show NAD⁺/NADH ratio in their 2DG perturbation in mouse. Ideally, it would be good to see the NAD⁺/NADH ratio in 1mM 2DG + 1mM azide perturbation as well since the combination shows a global perturbation of Hes7 period similar to changes in temperature. Moreover, Supp Fig 5 indicates a PER comparable between 2DG+azide treatment and controls. Instead, 2DG only treatment reduces PER. This may imply recovery of glycolytic function in the 2DG+azide and suggests that when both pathways are inhibited the cells adjust to increase PER/redox. This may have implications in terms of global effects of energy consumption/redox in the period of Hes7: this treatment rewires the energetic use of cells potentially towards glycolysis which the authors should discuss. Have they considered consecutive addition of 2DG followed by azide treatments instead of combinatorial?

2-1. Following the suggestion, we measured the NAD⁺/NADH redox balance in mouse PSM cells treated with several metabolic inhibitors, including the combination of 2DG and azide (new Fig. 3h). Glycolysis inhibitor 2DG increased the NAD⁺/NADH ratio while ETC inhibitor azide decreased it, as expected. However, there was no correlation between the NAD⁺/NADH ratio and the segmentation clock period across five metabolic inhibition conditions. Thus, we cannot say that the NAD⁺/NADH is the single important metabolic factor dictating the clock period.

Regarding the comparison between 2DG only and 2DG+azide treatments (new Supplementary Fig. 7, this should be a misunderstanding. The dosages of 2DG are different between the two conditions (10 mM 2DG only vs. 1 mM 2DG+azide), and 2DG-only treatment caused a greater PER reduction because of the higher dosage. Namely, azide did not buffer the effects of 2DG. We chose 10 mM 2DG and 1 mM 2DG+azide just because we wanted to assess the metabolic rates in the conditions that had actually extended the segmentation clock period (new Fig. 2c and 3b, respectively).

> Given that the NAD⁺/NADH ratio parameter has been found to be upstream of Hes7 oscillations, it would be good to understand how they think its perturbation affects Hes7 production delay and splicing. According to Diaz-Cuadros et al., we would not expect to see an effect on Hes7 degradation.

2-2. As discussed in point 2-1, the NAD⁺/NADH ratio, previously identified as the downstream target of azide, did not correlate with the segmentation clock period across various metabolic inhibition conditions (new Fig. 3h). This result leaves the mechanism by which azide modulates the intron delay unclear. 2DG may influence glycolytic metabolites and the pentose phosphate pathway, modulating the production delay and protein degradation. We discussed that “specific metabolites and metabolic factors that are regulated by distinct metabolic inhibitors, such as glycolytic metabolites and pentoses, may modulate the individual processes”.

Regarding protein degradation, while Diaz-Cuadros et al. demonstrated that azide treatment does not slow down protein degradation, which is consistent with our Hes7 degradation data, they did not examine the effects of 2DG treatment. Additionally, while Diaz-Cuadros et al. demonstrated that treatment with proteasome inhibitors does not extend the segmentation clock period, they assessed the bulk protein degradation rate only, leaving it unclear whether Hes7 protein degradation was slowed down or not.

2. The titration and characterization for 2DG, azide and oligomycin treatments is very neat. Oscillations in 2DG treatment or azide+2DG are dampened and quite hard to see in comparison to the controls. What quality controls were performed to determine the viability of the cells before/after treatment? It would be important to understand if control and treated cells are both as viable for the comparisons.

2-3. This is an important point. We agree that the samples treated with metabolic inhibitors, especially 2DG and oligomycin, show low signals of the luciferase reporter. Even in those samples, however, the oscillations are relatively clear (see the single tracks of raw oscillation data in new Supplementary Figs. 1b; 6a-c; 11a,b). These oscillations were less visible in the averaged signals of the original figures (now new Supplementary Figs. 1a; 6a-c; 11a,b) mostly because the Y-axis was adjusted to match the high signals of control samples.

In some samples, however, the oscillations indeed begin to dampen after ~10 hours. To assess the viability of cells, we took bright-field images at 0, 7, and 21 hours after several metabolic inhibitor treatments (new Supplementary Fig. 1c). Cells seemed relatively healthy at 7 hours. At 21 hours, however, cells under harsh metabolic inhibitions, such as 2DG+azide and 2DG+oligomycin, showed much lower densities and more floating cells than the control, suggesting significant cell death.

Based on this quality control experiment, we concluded that measurements within 10 hours are not severely impacted by cell death. The degradation rate and production delay assays are quick enough. In contrast, we should be careful about the oscillation assays for the period and intron delay quantifications. Luckily, even though the oscillation sometimes began to dampen after 10 hours, the periods per se were relatively consistent across the entire time course.

3. All treatments to measure Hes7 parameters and the Seahorse measurements are done in medium supplemented with glucose, pyruvate and glutamine. Are the concentrations in the medium and Seahorse medium for these parameters the same? If they aren't, I wonder to what extent the responses measured for Hes7 period parameters and the Seahorse are comparable. Would we expect them to be affected by the default concentrations?

2-4. We employed comparable concentrations of glucose, pyruvate, and glutamine between the Hes7 parameter and Seahorse measurements. More specifically, the following concentrations were used for the parameter and Seahorse measurements, respectively: glucose 17.5 mM and 20 mM; pyruvate 1 mM and 1 mM; glutamine 2.5 mM and 2 mM. It is true, however, that the base media are different between these two measurements because the Seahorse assay requires a specific medium, and we added a warning: "Note that this medium composition differs from the one used for the other kinetic parameter measurements".

4. The authors conclude that metabolic activities may not act as global modulators for the segmentation clock tempo. However, the combinatorial addition of glycolytic and ETC inhibitors (1mM 2DG + 1mM azide and/or oligomycin) in mouse PSM cells affects the three parameters of Hes7. To me, this result somewhat opposes their interpretation as I can imagine a scenario where the combined treatment with inhibitors mimics a global endogenous regulator (enzyme/metabolite) that rewires energy consumption in cells to increase solely PER. This perturbation is to me the most interesting/puzzling metabolic effect. It is also a complicated experiment to put a lot of emphasis on because the oscillations are quite damped, but I think that some additional info and tests would strengthen the findings:

> As mentioned earlier, they observe that the inhibition reduces OCR but not glycoATP, and state that they were expecting a synergistic effect on energy consumption. How do the authors think that the cells are buffering the response to 2DG when they add azide? Do they observe a similar energetic phenotype on the Seahorse if they treat cells with 2DG and Oligomycin?

2-5. We are sorry that our intention was not clear enough. We meant that 'individual' metabolic activities are selective while 'combinations' of multiple metabolic activities can be global. We slightly modified the wording to highlight the difference between individual and combined metabolic activities, discussing this topic in the following paragraphs:

"It is noteworthy that the combination of glycolysis and ETC inhibitions affected the three key molecular processes simultaneously (1 mM 2DG + 1 mM azide in Fig. 4), fulfilling the criteria as a global modulator for the segmentation clock tempo."

"Although metabolism had been an attractive candidate for a global modulator of the segmentation clock tempo due to the universal role of energy, individual metabolic inhibitions were not as universal as temperature change which affected all three processes simultaneously. Instead, the combination of glycolysis and ETC inhibitions affected the three key molecular processes, synergistically extending the segmentation clock period."

"As any single metabolic activities examined in this study are not a global modulator for the three key molecular processes of the segmentation clock".

As additional notes, azide does not buffer the effect of 2DG (discussed in point 2-1), and oscillations are relatively stable for ~10 hours (discussed in point 2-3).

> How do they interpret the difference in the results obtained in 2DG+Oligomycin versus 2DG+Azide for intron delay and production delay?

2-6. Regarding the intron delay, azide treatment alone extended it while oligomycin did not. So, the results of 2DG+azide and 2DG+oligomycin are reasonable. Regarding the production delay, in contrast, we cannot clearly explain why 2DG+azide and 2DG+oligomycin exhibit distinct effects while either azide or oligomycin alone did not influence the production delay.

> How would these perturbations look in human?

2-7. To address the question, we treated human PSM cells with both 2DG and azide (new Fig. 5c,d; new Supplementary Fig. 11b). The combined treatment synergistically extended the human segmentation clock period, consistent with the results in mouse cells.

> Given that the perturbation seems to have a global effect for Hes7, do they observe a change in the overall speed of differentiation beyond Hes7?

2-8. Our study focuses on determining whether metabolic activities act as a global modulator of the three key molecular processes of Hes7. While examining effects on other genes falls outside the scope of this work, we recently demonstrated the impact of the glycolysis inhibitor 2DG on broader protein degradation rates using quantitative mass spectrometry (Matsuda et al, bioRxiv 2024, <https://doi.org/10.1101/2024.06.07.597977>). This led us to state, “We recently observed this decelerating effect of 2DG on the degradation of numerous other proteins as a consistent trend”. Assessing the effects on the overall speed of differentiation is our next challenge.

5. Production delay serves as a proxy for “a combined delay caused by the gene expression steps of Hes7, including transcription and translation, except for intron-related steps”. I wonder if the authors could unpick whether the perturbations they measure affect more transcription or translation?

2-9. While we understand the reviewer’s point, further decomposing the production delay is technically challenging. Our group is currently working on establishing single-molecule fluorescent imaging to measure the individual steps of production, but this will require considerable time to develop.

6. The authors show that perturbation temperature affects Hes7 parameters and conclude that temperature acts as a global regulator for tempo. While cells from various species grown at similar temperatures in vitro show differences in Hes7 period, this perturbation may be insightful to understand changes in the overall rate of tempo. At present the association between the period of Hes7 and the rate of development is still correlative. What other phenotypic effects do the authors observe in cells? And at higher temperatures? It would be good to verify that shifts in temperature indeed affect changes in tempo globally and not just to Hes7.

2-10. To assess a more global tempo, we checked the onset timing of Hes7 expression, instead of Hes7 oscillation. Namely, Hes7 induction was used as a proxy for PSM cell differentiation tempo. At 30 °C, the Hes7 induction timing was indeed delayed by 500-800 min (see Supporting Data 1). Although this is a

promising research direction, our assessment of cell differentiation timing is preliminary, and we have chosen not to include it in the main manuscript. Additionally, increasing the temperature of human cell cultures by 7 °C would impose much higher stress on cells than decreasing it.

In reply to Reviewer 3:

Matsuda et al. investigate how specific metabolic interventions affect the kinetics of the key molecular processes that determine the segmentation clock period. They employ previously established assays to measure the duration of protein half-lives, protein production and intron delay of the Hes7 gene product together with the segmentation clock period upon perturbations of glycolysis, the electron transport chain and ATP production. All perturbations extended the clock period but affected different molecular processes underlying the clock period. Although the mechanistic basis of the linkage between individual perturbations and the kinetics of specific molecular processes has not been investigated, this is a new and unexpected finding and an important contribution to the field of developmental timing as it shows how timing can be modulated through different metabolic and molecular processes. Finally, the authors show that combined perturbations exhibited partially synergistic effects, that the selective effects of metabolic perturbations are shared between mouse and human, and that the effect of temperature on clock period emerges from changes in the kinetics of all underlying molecular processes.

The manuscript has a clear structure and is easy to follow, so are the figures.

Specific points

Major

1. One major concern with the manuscript in its current form is that the authors do not show any primary data of the cells under the different perturbations. One would expect that – depending on the concentrations used - the inhibitors have generic effects on cell viability. For the reader to evaluate how well cells tolerate the perturbations, the authors should show cell images or movies of at least a few representative cases, e.g. for the experiment shown in Supplementary Figure 4 where inhibitor treatments lowered the Hes7 reporter signal drastically. How well can peaks be identified and how long do cells keep oscillating under these conditions? It would also be interesting to know how cell proliferation is affected by the different perturbations.

3-1. This is an important point. As primary data, we have added the single tracks of raw oscillation data and the bright-field images of cells.

We agree that the samples treated with metabolic inhibitors, especially 2DG and oligomycin, show low signals of the luciferase reporter. Even in those samples, however, the oscillations are relatively clear (see the single tracks of raw oscillation data in new Supplementary Figs. 1b; 6a-c; 11a,b). These oscillations were less visible in the averaged signals of the original figures (now new Supplementary Figs. 1a; 6a-c; 11a,b) mostly because the Y-axis was adjusted to match the high signals of control samples.

In some samples, however, the oscillations indeed begin to dampen after ~10 hours. To assess the viability of cells, we took bright-field images at 0, 7, and 21 hours after several metabolic inhibitor treatments (new Supplementary Fig. 1c). Cells seemed relatively healthy at 7 hours. At 21 hours, however, cells under harsh metabolic inhibitions, such as 2DG+azide and 2DG+oligomycin, showed much lower densities and more floating cells than the control, suggesting significant cell death.

Based on this quality control experiment, we concluded that measurements within 10 hours are not severely impacted by cell death. The degradation rate and production delay assays are quick enough. In contrast, we should be careful about the oscillation assays for the period and intron delay quantifications. Luckily, even

though the oscillation sometimes began to dampen after 10 hours, the periods per se were relatively consistent across the entire time course.

2. Figure 4b: Can the authors explain why 1 mM 2DG and 1 mM azide separately do not have an effect on protein half-life but together they increase half-life? Figure 2e shows that 10 mM 2DG increases half-life to a similar degree as 1 mM 2DG and 1 mM azide combined. Could the combined effect result from a generally lower metabolism compared to the single inhibitor treatment that is also seen when increasing the 2DG dose to 10 mM?

3-2. It is true that we currently lack an explanation for the mechanism of the synergistic effect of 1 mM 2DG and 1 mM azide on protein degradation. Still, we would not attribute the synergy to a generally lower metabolism because the effect of this combined metabolic inhibition is selective – 2DG+azide did not show a synergy on the intron delay or metabolic rates (new Figs. 4e, 3g).

3. Are the changes to the kinetics of specific molecular processes sufficient to explain the changes in the clock period? In a previous publication (PMID 32943519), the authors used a mathematical model to study the dependency of the segmentation clock period on the kinetics of different biochemical reactions. Using this model to test how well the inhibitor treatments explain changes in the clock period would give the manuscript a significant lift.

3-3. Following the suggestion, we simulated the oscillation periods using the measured three kinetic parameters (new Supplementary Fig. 10c). The simulation successfully reproduced the ranking of period extension across the 2DG (10 mM), azide (1 mM), and 2DG (1 mM) + azide (1 mM) conditions. Although the 2DG (1 mM) + oligomycin (10 nM) condition showed a clear discrepancy between the simulation and experiment, this is probably because this simulation resulted in a damped oscillation. Additionally, our simulation tended to underestimate the contribution of Hes7 protein degradation rate to the period extension.

Minor

4. Regarding the intron delay assay, please explain the constructs in more detail, e.g. in the methods section or figure legend. Do the constructs contain any Hes7 coding sequence and could therefore feed back on the clock oscillations? What is the length, identity and position of the introns in the intron-containing construct?

3-4. We are sorry that our explanation was not clear enough. As the constructs are complicated, we have included an additional schematic (new Supplementary Fig. 3a) and cited our previous paper where the actual sequences are available. Although the constructs contain Hes7 coding sequence, Hes7 protein is not translated due to the stop codon immediately before the Hes7 gene. Thus, this construct should not feed back on oscillations.

5. In Supplementary Figure 1 (generally in the supplementary figures), do the three plots per condition represent replicates? If so, please state in figure legend.

3-5. They are indeed three biological replicates. We labeled them in all relevant supplementary figures.

In reply to Reviewer 4:

We thank all the reviewers for their constructive comments.